# New insights into the polar ozone and water vapor, radiative effects, and their connection to the tides in the mesosphere-lower thermosphere during major Sudden Stratospheric Warming events

Guochun Shi[1,2], Hanli Liu[3], Masaki Tsutsumi[4,5], Njål Gulbrandsen[6], Alexander Kozlovsky[7], Dimitry Pokhotelov[8], Mark Lester[9], Christoph Jacobi[10], Kun Wu[3,*], and Gunter Stober[1,2]

[1]Oeschger Center for Climate Change Research, University of Bern, Bern, Switzerland
[2]Institute of Applied Physics, University of Bern, Bern, Switzerland
[3]High Altitude Observatory, National Center for Atmospheric Research, Boulder, CO, USA
[4]National Institute of Polar Research, Tachikawa, Japan
[5]The Graduate University for Advanced Studies (SOKENDAI), Tokyo, Japan
[6]Tromsø Geophysical Observatory UiT - The Arctic University of Norway, Tromsø, Norway
[7]Sodankylä Geophysical Observatory, University of Oulu, Finland
[8]Potsdam Institute for Climate Impact Research, Member of the Leibniz Association, Potsdam, Germany
[9]Department of Physics and Astronomy, University of Leicester, Leicester, UK
[10]Institute for Meteorology, Leipzig University, Leipzig, Germany
[*]Now: School of Physics and Electronic Sciences, Changsha University of Science and Technology, Changsha, China

**Correspondence:** Guochun Shi (guochun.shi@unibe.ch)

**Abstract.** We examine the variability of diurnal (DT), semidiurnal (SDT), and terdiurnal (TDT) tide amplitudes in the Arctic mesosphere and lower thermosphere (MLT) during and after sudden stratospheric warming (SSW) events using meteor radar data at three polar-latitude stations: Sodankylä (67.37°N, 26.63°E), Tromsø (69.58°N, 19.22°E), and Svalbard (78.99°N, 15.99°E) as well as one station outside the polar vortex located at Collm (51.3°N, 13°E). By combining tidal amplitude anomalies with trace gas variations, induced by large-scale dynamical changes caused by the breaking of planetary waves, this study provides new observational insights into the variation of ozone and water vapor, transport, and tides at polar latitudes. We use short-wave (QRS) and long-wave (QRL) radiative heating and cooling rates simulated by the WACCM-X(SD) model to investigate the roles of polar ozone and water vapor in driving mesospheric tidal variability during SSWs in the polar regions. Our analysis reveals distinct tidal responses during SSW events. At the onset of SSWs, a significant negative anomaly in TDT amplitudes in zonal and meridional components is observed, with a decrease of 3 m/s, approximately 25% change compared to the mean TDT amplitude. Meanwhile, SDT shows a positive anomaly of 10 m/s, with changes reaching up to 40% ,indicating an enhancement of tidal amplitude in both components. The DT amplitude exhibits a delayed enhancement, with a positive amplitude anomaly of up to 5 m/s in the meridional wind component, occurring approximately 20 days after the onset of SSWs. A similar, but weaker effect is observed in the zonal wind component, with changes reaching up to 30% in the zonal component and 50% in the meridional wind component. We analyzed the contributions of ozone and water vapor to the short-wave heating and long-wave cooling before, during, and after the onset of SSW events. Our findings suggest that the immediate responses of SDT are most likely driven by dynamical effects accompanied by the radiative effects from ozone. Radiative forcing change

during SSW likely plays a secondary role in DT tidal changes, but appears to be important 20 days after the event, particularly during the spring transition. Water vapor acts as a dynamical tracer in the stratosphere and mesosphere but has minimal radiative forcing, resulting in a negligible impact on tidal changes. This study presents the first comprehensive analysis of mesospheric tidal variability in polar regions during sudden stratospheric warmings (SSWs), examining and linking the significant role of trace gases and radiative effects in modulating tidal dynamics.

## 1 Introduction

Major SSW events are dramatic disruptions of the winter polar stratosphere, characterized by rapid temperature increases and the reversal of the typical westerly winds. These events occur due to the interaction between planetary waves propagating from the troposphere into the stratosphere and the stratospheric mean circulation (Matsuno, 1971; Andrews et al., 1987). This interaction leads to the weakening or splitting of the winter stratospheric polar vortex (Haynes et al., 1991; Matthias et al., 2013), resulting in a circulation reversal, increased downwelling in the polar stratosphere, and a subsequent rise in temperature due to adiabatic heating. SSWs have broad impacts, influencing surface temperature (Davis et al., 2022; Hall et al., 2021), weather patterns in the troposphere (Baldwin et al., 2021; Domeisen et al., 2020), large-scale circulation (Iida et al., 2014), stratospheric transport and composition (de la Cámara et al., 2018; Schranz et al., 2020; Shi et al., 2024), and altering the behavior of atmospheric tides in the MLT regions (Becker, 2017; Zhang et al., 2021; Liu et al., 2022) and up to the ionosphere (Fang et al., 2012; Pedatella and Liu, 2013; Jones Jr. et al., 2020; Günzkofer et al., 2022).

Several observational and numerical studies have established that the occurrence of SSW events influences the tidal variabilities in the MLT across equatorial latitudes (Sridharan et al., 2009; Lima et al., 2012; Jin et al., 2012; Sathishkumar and Sridharan, 2013; Siddiqui et al., 2018; Liu et al., 2021), as well as middle and polar latitudes (Jacobi et al., 1999; Bhattacharya et al., 2004; Hoffmann et al., 2007; Pedatella et al., 2014; Chau et al., 2015; Stober et al., 2020; Liu et al., 2021; Eswaraiah et al., 2018; Hibbins et al., 2019; Zhang et al., 2021; Dempsey et al., 2021; Liu et al., 2022; van Caspel et al., 2023; Dutta et al., 2024). For instance, van Caspel et al. (2023) used a mechanistic tidal model to investigate the response of SDT to the 2013 SSW event and compared their findings with Meteor Radar (MR) wind observations at three stations: CMOR (43.3°N, 80.8°W), Collm (51.3°N, 13.0°E), and Kiruna (67.5°N, 20.1°E). Hibbins et al. (2019) observed an enhancement in the mid-latitude migrating SDT in the MLT regions around 10–17 days after the SSW onset using meteor wind data from the Super Dual Auroral Radar Network (SuperDARN) in the Northern Hemisphere. Dutta et al. (2024) reported an increase in solar SDT amplitude in the polar MLT regions during the boreal SSW event of 2013 and the austral SSW event of 2019. Additionally, Sathishkumar and Sridharan (2013) found a significant enhancement of DT amplitude in the zonal wind and the strength of the equatorial electrojet just before the onset of SSW, with the solar SDT dominating over the DT during the SSW.

Previous studies have suggested that the primary mechanisms driving SDT variability in the MLT during SSW include the modification of tidal amplitudes via propagation of tidal waves (Jin et al., 2012; Eswaraiah et al., 2018; He and Chau, 2019; van Caspel et al., 2023), and the nonlinear interaction between tides and planetary waves (Liu et al., 2010b; Pedatella and Forbes, 2010; Lima et al., 2012; He et al., 2020, 2024). For instance, Lima et al. (2012) demonstrated that the intensified

tides and quasi-two-day wave amplitudes at low-latitude observed during a major SSW event are associated with strengthened planetary wave activity in the stratospheric winter of the Northern Hemisphere. Given that the absorption of solar ultraviolet radiation (UV) by stratospheric ozone is the primary source of SDT (Forbes and Garrett, 1978; Lindzen and Chapman, 1969), and that SSWs affect the distribution of stratospheric and mesospheric ozone, changes in ozone density could potentially

influence the enhancement of SDT during SSW (Goncharenko et al., 2012; Pedatella et al., 2014; Limpasuvan et al., 2016; Eswaraiah et al., 2018; Stober et al., 2020; van Caspel et al., 2023). Goncharenko et al. (2012) reported that the prolonged increase in tropical ozone density around peak ozone heating rates generated a migrating semidiurnal tide, while circulation changes amplified longitudinal inhomogeneities in ozone distribution, potentially leading to the generation of non-migrating tides. Limpasuvan et al. (2016) suggested that the migrating SDT is globally amplified during the 20–30 day interval following

SSW onset, likely due to enhanced stratospheric ozone in the tropics and associated solar heating linked to equatorial upwelling and cooling caused by the SSW. Eswaraiah et al. (2018) studied ozone transport over Antarctica and explored the nonlinear interaction between planetary waves and tides to understand tidal enhancement observed 3 to 4 weeks after the central day of SSWs. Moreover, Siddiqui et al. (2019) utilized WACCM simulations to investigate tidal amplitudes during the 2009 SSW event, highlighting the crucial role of stratospheric ozone variability in modulating semidiurnal solar tidal changes. While the

nonlinear interaction between tides and planetary waves is considered a primary cause of SDT enhancement in the MLT region, the impact of stratospheric ozone and water vapor on tidal changes during SSW events has been studied more extensively in tropical regions than in polar regions (Sridharan et al., 2012; Goncharenko et al., 2012; Pedatella and Liu, 2013; Siddiqui et al., 2019). Motivated by the observed links between trace gas variations in tropical regions during SSWs and changes in wave-tidal amplitudes in the MLT, this study aims to explore how trace gases, specifically ozone and water vapor, change in polar regions

and their potential influence on tidal amplitudes during SSW events.

This study examines the causes of mesospheric tidal variability in the polar regions during SSWs, specifically focusing on the role of radiative effects from ozone and water vapor. We use long-term MLT wind measurements from MRs at northern polar-latitude stations: Sodankylä (67.37°N, 26.63°E), Tromsø (69.58°N, 19.22°E), and Svalbard (78.99°N, 15.99°E) to analyze the variability of DT, SDT, and TDT in the zonal and meridional wind components compositing 10 major SSW events from 2004 to

75 2023 and the Collm (51.3°N, 13°E) MR outside the polar vortex as a reference (see in Appendix B). We utilize the short-wave (QRS) heating and long-wave (QRL) cooling rates simulated by the Specified Dynamics (SD) Whole Atmosphere Community Climate Model with thermosphere and ionosphere extension (WACCM-X) to investigate heating and cooling rates associated with ozone and water vapor responses to SSW concerning tidal variations in the MLT region. In the first step, we compare water vapor and ozone anomalies from WACCM-X(SD) with observations to ensure consistency with the observational data

before analyzing their role in tidal variability. This study presents a quantification of total radiative forcing changes during SSW events and their close correspondence with ozone and water vapor changes observed at polar latitudes. Previous studies already compared WACCM-X(SD) tides and mean MR winds together with other general circulation models (GCMs) (Stober et al., 2021b). The combined analysis of tidal amplitude anomalies and trace gas variations in the polar regions provides new insights into the factors influencing tidal dynamics during SSWs.

## 2    Data and Methodology

### 2.1    Meteor radar data and analysis

Meteor radar observations collected at three stations in high latitudes are located at Svalbard (78.99°N, 15.99°E, from March 2001 to present), Tromsø (69.58°N, 19.22°E, from November 2003 to present), and Sodankylä (67.37°N, 26.63°E, from December 2008 to present) in the Arctic, and Collm (51.3°N, 13°E, from August 2004 to present) in the mid-latitude regions. All systems were almost continuously in operation for measuring zonal and meridional winds in the MLT region with a temporal resolution of 1 h and vertical resolution of 2 km, which uses the same wind retrieval algorithm (Stober et al., 2021a, b). The wind retrieval algorithm is a further development of the wind analysis introduced by Hocking et al. (2001) and Holdsworth et al. (2004). The total tidal amplitude and phases are estimated using the adaptive spectral filter (ASF2D) (Baumgarten and Stober, 2019; Stober et al., 2020; Krochin et al., 2024). The total tides, usually dominated by migrating (DW1, SW2, TW3) tidal components, are obtained using the following function:

$$T(t,z), u(t,z), v(t,z) = T_0(z), u_0(z), v_0(z) + \sum_{n=1}^{3} \left[ A_n(z) \sin\left(\frac{2\pi}{P_n}t\right) + B_n(z) \cos\left(\frac{2\pi}{P_n}t\right) \right], \tag{1}$$

where T, u, and v are the temperature, zonal, and meridional winds, respectively. $P_n$ is 8, 12 and 24 h, corresponding to terdi-urnal tides (TDT), semidiurnal tides (SDT) and diurnal tides (DT), respectively. $A_n$ and $B_n$ denote the Fourier coefficients for the tidal amplitudes at a given altitude. These coefficients are regularized, assuming certain smoothness of the tidal phase with altitude. The zonal mean zonal and meridional wind and the zonal mean temperature are given by $T_0$, $u_0$, and $v_0$, respectively. The retrieval function also includes longer period waves such as the quasi-two-day wave (QTDW) and stationary planetary waves (Baumgarten and Stober, 2019; Schranz et al., 2020).

### 2.2    GROMOS-C

GROMOS-C (GRound-based Ozone MOnitoring System for Campaigns) is an ozone microwave radiometer that measures the ozone emission line at 110.836 GHz at Ny-Ålesund, Svalbard (78.99° N, 12° E) since September 2015. It was built by the Institute of Applied Physics at the University of Bern (Fernandez et al., 2015). Measured ozone profiles are retrieved from the ozone spectra with a temporal averaging of 2 hours, leveraging the Atmospheric Radiative Transfer Simulator version-2 (ARTS2; Eriksson et al., 2011) and Qpack2 software (Eriksson et al., 2005) according to the optimal estimation algorithm (Rodgers, 2000). The retrieved ozone profile has a vertical resolution of 10-12 km in the stratosphere and up to 20 km in the mesosphere, covering an altitude range from 23 to 70 km. The measured datasets were used to study the photochemically induced diurnal cycle of ozone in the stratosphere and lower mesosphere (Schranz et al., 2018). The ozone measurements of GROMOS-C have been validated with AURA-MLS and MERRA-2 (Schranz et al., 2020; Shi et al., 2023, 2024). Furthermore, GROMOS-C has proved capable of measuring the tertiary ozone layer above Ny-Ålesund, Svalbard, in winter (Schranz et al., 2018).

## 2.3 MIAWARA-C

MIAWARA-C (MIddle Atmospheric WAter vapor RAdiometer for Campaigns) is a ground-based microwave radiometer measuring the pressure-broadened rotational emission line of water vapor at the frequency of 22 GHz. The University of Bern built this instrument (Straub et al., 2010) and performed a campaign at Ny-Ålesund, Svalbard (78.99° N, 12° E) since September 2015. MIAWARA-C retrieval, like GROMOS-C, is conducted using the ARTS2 (Eriksson et al., 2011) and QPACK2 software (Eriksson et al., 2005), following the optimal estimation algorithm (Rodgers, 2000). From the measured spectra, the retrieved water vapor profiles cover an altitude range extending from 37 km to 75 km with a time resolution of 2-4 h and a vertical resolution of 12-19 km. MIAWARA-C measurements were validated against MERRA-2 reanalysis, MLS observations, and WACCM simulations, followed by a comprehensive intercomparison (Schranz et al., 2019, 2020; Shi et al., 2023). Moreover, the effective ascent and descent rates of air were estimated using the water vapor from MIAWARA-C as a passive tracer to investigate the dynamics of transport processes in the Arctic middle atmosphere (Straub et al., 2010; Schranz et al., 2019; Shi et al., 2023).

## 2.4 Aura-MLS

NASA's Earth Observing System (EOS) Microwave Limb Sounder (MLS) instruments on board the Aura spacecraft measure thermal emissions from the limb of Earth's atmosphere. MLS provides comprehensive measurements of vertical profiles of temperature and 15 chemical species from the upper troposphere to the mesosphere, spanning nearly pole-to-pole coverage from 82°S to 82°N (Waters et al., 2006; Schwartz et al., 2008). Aura MLS version 5 Level 2 profile measurements of ozone and water vapor volume mixing ratios (VMR) between August 2004 and December 2022 are used in this study. The pressure range for MLS ozone measurements useful for scientific applications extends from 261 to 0.0215 hPa (16-86km), while for water vapor it ranges from 316 to 0.00215 hPa (10-86km). The MLS water vapor dataset has been compared globally with ground-based microwave radiometers, typically showing values that are 0–10% higher than the profiles obtained from the microwave radiometers in the range of 3–0.03 hPa (Nedoluha et al., 2017). The ozone profiles from MLS and ground-based microwave radiometer measurements agree within 5% in the range of 18–0.04 hPa (Boyd et al., 2007; Bell et al., 2024). Relative differences of ozone and water vapor climatologies at polar stations from Aura-MLS and radiometers agree well, with relative differences mainly within ±7% throughout the middle and upper stratosphere (Shi et al., 2023). In this study, ozone and water vapor profiles are extracted for locations within ±1.2° latitude and ±6° longitude of Ny-Ålesund, Svalbard, Sodankylä, and Tromsø.

## 2.5 WACCM-X

The Whole Atmosphere Community Climate Model with thermosphere and ionosphere extension (WACCM-X) is an atmospheric configuration of the NCAR's Community Earth System Model (CESM) that extends into the thermosphere with a model top boundary between 500 and 700 km (Liu et al., 2018). WACCM-X can be run with coupled or prescribed ocean, sea ice, and land components, enabling studies at all atmospheric levels, including thermospheric and ionospheric weather

and climate. Physical processes represented in WACCM-X are built upon those in the regular WACCM configuration, which has a model top at 145 km, which in turn is built upon the Community Atmosphere Model (CAM) with its top at the lower stratosphere. The physics of these models is described in Marsh et al. (2013); Gettelman et al. (2019) and Neale et al. (2013).
WACCM-X includes an interactive chemistry module that describes the major chemical processes, including ozone and related chemical tracers. Radiative transfer calculations in WACCM-X are based upon those in WACCM3 and described in Liu et al. (2010a).

We performed a climatological run from 2015 to 2023, leveraging the well-established Specified Dynamics mode of the model (WACCM-X(SD)) setup corresponding to the GROMOS-C and MIAWARA-C campaigns. WACCM-X(SD) is a version
of WACCM-X whose temperature and winds from the surface to the stratosphere at $\sim 50$ km are constrained by the Modern-Era Retrospective Analysis for Research and Applications version 2 (MERRA-2) reanalysis dataset (Gelaro et al., 2017). By nudging with MERRA-2, the model states correspond closely to the actual meteorological state up to the highest nudged altitudes close to the stratopause. The model outputs (winds, temperature, trace gases, QRL, and QRS cooling/heating rates) have a conventional latitude-longitude grid with a horizontal resolution of $1.9\,^{\circ} \times 2.5°$ and a time resolution of 3 hours.
The vertical resolution is the same as WACCM below 0.96 hPa but has been increased to one-quarter scale height above that pressure level. The model top pressure is $4.1 \times 10^{-10}$ hPa (typically between 500 and 700 km altitude, depending on the solar and geomagnetic activity). The QRS heating rate is primarily governed by ozone and water vapor absorption of radiation at stratospheric altitudes. In the mesosphere, the QRS is affected by atomic oxygen and direct exothermic heating as well as other ionospheric energy sources, such as particle precipitation. In contrast, the presence of carbon dioxide, water vapor, and ozone
influences the QRL cooling rate.

## 2.6 MERRA-2

The Modern-Era Retrospective Analysis for Research and Applications, version 2 (MERRA-2) is a global atmospheric reanalysis produced by NASA's Global Modeling and Assimilation Office (GMAO), using the Goddard Earth Observing System-5 (GEOS-5) atmospheric general circulation model. Covering the period from 1980 to the present, MERRA-2 assimilates a wide
range of observational datasets to generate three-dimensional (a horizontal resolution of $0.5° \times 0.625°$ and 72 model levels from the surface to 0.01 hPa) meteorological fields (e.g. zonal and meridional winds, temperature, ozone, and water vapor) with a 3-hourly time resolution. This study uses zonal and meridional wind fields for comparison with WACCM-X(SD) model simulations.

## 3 Results

### 3.1 Initial comparison of WACCM-X(SD) and combined Meteor Radar and MERRA-2 fields for SSW 2018/2019

We first evaluate the wind patterns observed by Meteor Radar (MR) + MERRA-2 and WACCM-X(SD) in the stratosphere, mesosphere, and lower thermosphere. Figure 1 illustrates the variability in zonal and meridional wind components above

Tromsø during the winter 2018/2019 SSW event, covering the altitudes from 20 to 100 km. The comparison shows that the combined MERRA-2 and MR winds agree reasonably well with WACCM-X(SD), reflecting even daily features. Zonal winds

agree within 2-5 m/s for the overlapping region at 75 km. Climatologically, mean meridional winds are generally weaker and around the zero line; thus, the agreement does not look as good, but the overall magnitude difference is similar within 2-5 m/s. MERRA2 and MR zonal and meridional winds often show very consistent tidal features and only occasionally some phase difference for individual days. WACCM-X(SD) winds are very similar to MERRA-2 up to the maximum nudging altitude, they start to generate features above that are neither reflected in MERRA-2 nor in the MR winds. These differences are likely due to

the decreasing nudging strength with altitude. Additionally, the gravity wave (GW) parameterizations may contribute to these discrepancies by driving the fields away from the observed state (Stober et al., 2021b). In particular, around the SSW there are strong zonal and meridional wind features that are not seen in the observations. We also noticed that the elevated stratopause does not extend beyond 75 km in WACCM-X(SD), but is found up to this height in the combined MR and MERRA-2 wind fields. The formation of the elevated stratopause is accompanied by the reformation of the polar vortex, indicated by the

eastward winds that start to intensify at the mesosphere ( 60–80 km) and gradually descend with time after the SSW event. The GW parameterizations tend to produce or strengthen certain tendencies in the wind fields, resulting in larger deviations of the instantaneous winds, while sustaining most of the large-scale wind patterns. This is in agreement with previous studies targeting the climatological behavior of MLT winds between MR winds and tides in comparison to WACCM-X(SD) (Stober et al., 2021b).

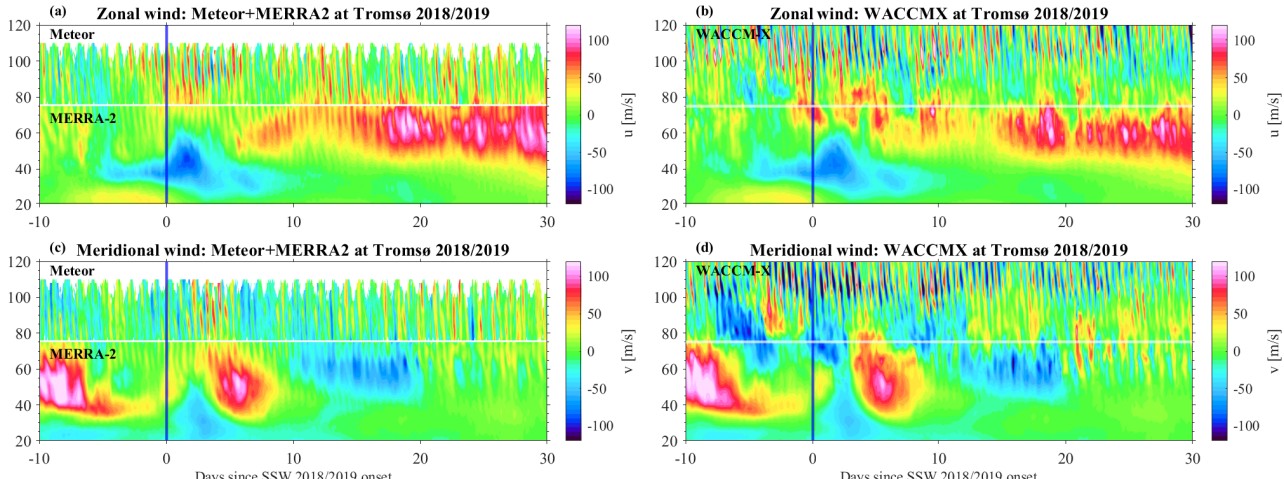

**Figure 1.** The cross-section of zonal and meridional wind variations at Tromsø (69.58°N, 19.22°E) during the SSW of 2018/2019, derived from (a, c) MR and MERRA-2, (b, d) WACCM-X(SD). The white horizontal line marks the 75 km altitude level. In the left panels, the horizontal line distinguishes meteor radar (upper section) from MERRA-2 (lower section). The vertical blue line denotes the central date of SSW on January 2, 2019.

Furthermore, we present and analyze WACCM-X(SD) fields that are not commonly used but provide essential information to investigate and understand the short-term tidal variability. Figure 2 shows the 3-hourly ozone and water vapor VMR as well as QRL and QRS for the location of Tromsø. The WACCM-X(SD) simulation reflects the exchange of airmasses in the middle atmosphere due to the SSW event. In particular, the sudden intrusion of water vapor into the mesosphere and the increase of ozone in the stratosphere are evident. The large-scale dynamics also directly affect the QRL and, thus, the cooling

of the middle atmosphere, resulting in a pronounced double-layer structure with increased cooling rates at around 40 km and above 80 km starting already 10 days before the SSW central day. These changes in the cooling rates are aligned with water vapor anomalies, but given the mean cooling rate of around 0.3 K/day due to water vapor in the mesosphere (Brasseur and Solomon, 2005), its contribution to the QRL cooling rate is negligible. Thus, variations in QRL are more closely aligned with temperature (warming in the stratosphere, cooling in the mesosphere, and warming above) rather than changes in water vapor.

For the short-term tidal variability, ozone and short-wave absorption might play a key role. WACCM-X(SD) shows the primary stratospheric ozone layer at 20-50 km, the tertiary ozone layer around 60-70 km, and the secondary ozone layer above 90 km. The secondary ozone layer reflects a strong diurnal modulation and starts to fade toward the spring transition. Furthermore, associated with the large intrusion and exchange of air masses, the secondary ozone layer is disturbed, which starts again about 10 days before the SSW event and restores afterward. These changes are accompanied by a strong response from the QRS.

Furthermore, the QRS also reveals a clear signature of a diurnal forcing starting 20-30 days after the SSW, which is related to the slowly descending elevated stratopause and the increased ozone VMR. During this time, after an SSW event, the secondary ozone layer at the MLT almost vanished. Although the 2018/19 SSW event can be considered to be representative of major SSWs, there is a noticeable variability between events. In section 3.2 and 3.3, we present composite analysis for major SSW events concerning the central day and focus on anomalies rather than absolute values.

We also performed a tidal analysis for the SSW 2018/19 event. Therefore, we applied the ASF2D to the MR and WACCM-X(SD) data for the location of Tromsø. Figure 3 shows the SDT anomalies for the zonal and meridional wind components, respectively. The anomaly is calculated by comparing each day's value to the climatological average for the same day across all years from 2015 to 2023. WACCM-X(SD) exhibits a much stronger semidiurnal tidal amplification than can be found in the observations for the horizontal wind. As already seen in the instantaneous winds, WACCM-X(SD) produces additional

features at the MLT, which likely also affect the tidal propagation, resulting in a decreased agreement when comparing daily features, while the overall behavior seems to still be reproduced. The comparison for the DT is showcased in Figure 4. The DT appears to be polarized and exhibits much stronger amplitudes in the meridional wind component compared to the zonal wind component. Furthermore, the MR observations as well as WACCM-X(SD) capture a diurnal tidal enhancement starting about 20 days after the SSW. However, the altitude range where this enhancement can be found is more confined below 80 km

in WACCM-X(SD), whereas the MR measurements indicate a strong DT up to 100 km. This difference seems to be related to the differences in the altitude coverage of the elevated stratopause presented in Figure 1.

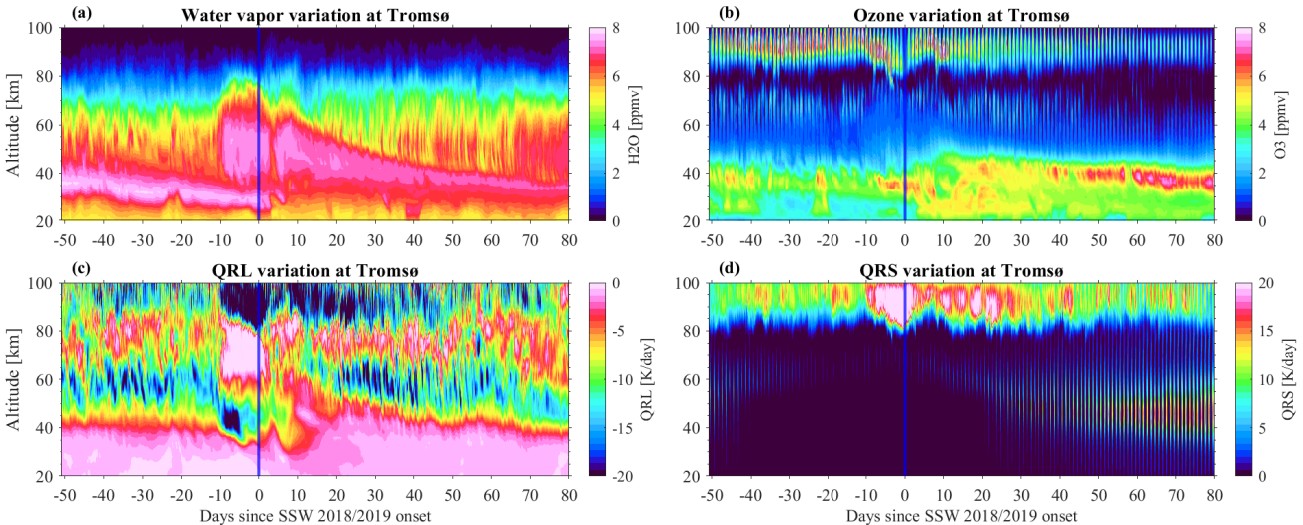

**Figure 2.** The cross-section of (a) water vapor, (b) ozone, (c) QRL cooling rate, and (d) QRS heating rate variations at Tromsø during the winter SSW of 2018/2019, derived from WACCM-X(SD).

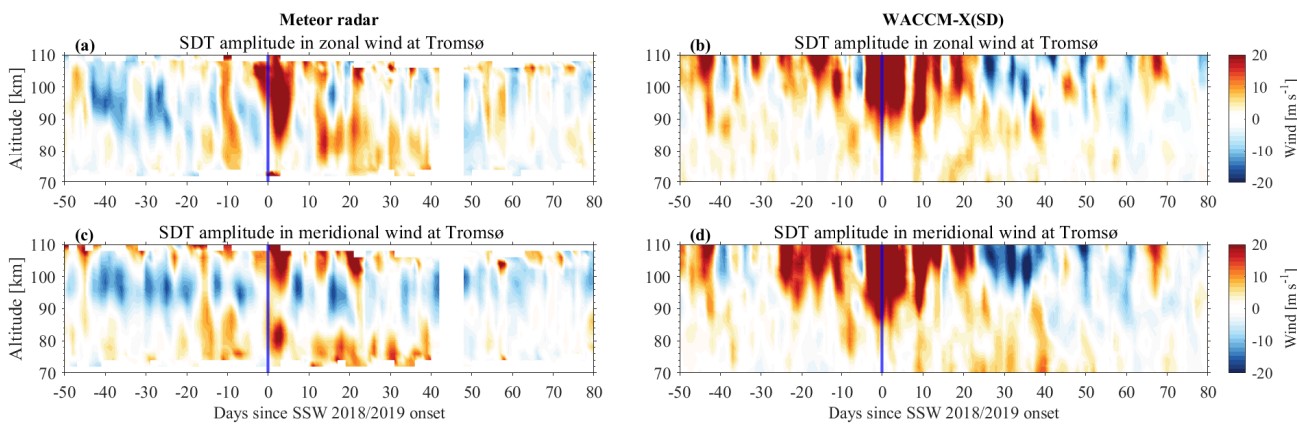

**Figure 3.** The cross-section of SDT amplitude anomalies in zonal and meridional wind components at Tromsø during the winter SSW of 2018/2019, derived from (a, c) MR and (b, d) WACCM-X(SD).

### 3.2 Composite analysis of mean wind response during SSWs

To understand the variation of winds and tides before and after SSW, the anomalies of zonal and meridional winds and tidal amplitude are investigated. Figure 5 presents the composite zonal and meridional wind anomalies observed by three MRs at Sodankylä, Tromsø, and Svalbard as a function of time relative to the SSW central date. The central date, as defined in Li et al. (2023) and Butler et al. (2017), corresponds to the onset of SSW, which is identified based on the zonal-mean zonal winds at

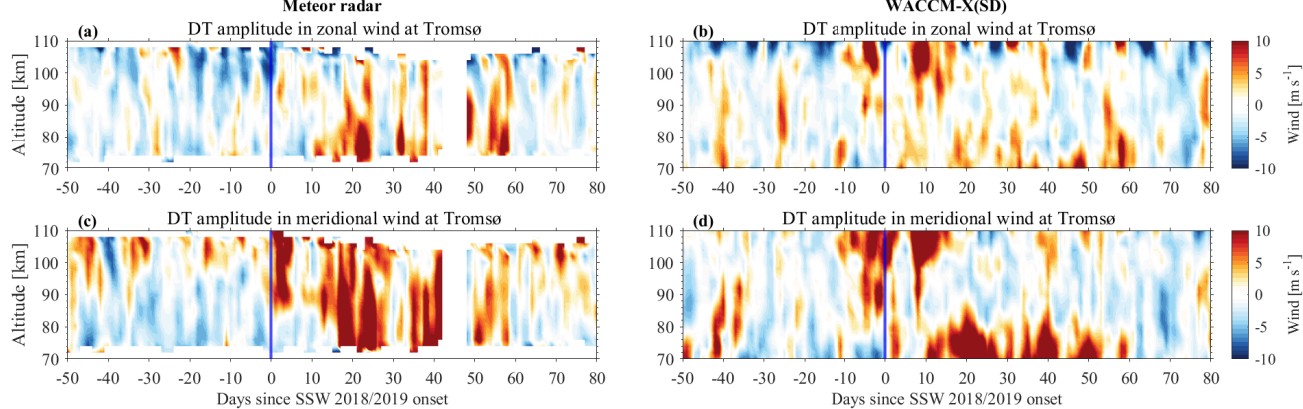

**Figure 4.** The same as Figure. 3 but for the DT amplitude.

60°N and 10 hPa reversing from westerly to easterly. 10 major SSW events from 2004 to 2023 are analyzed in this study (in Table A1). In the left panel of Figure 5, a reversal of the zonal wind is evident starting about 10 days before the nominal SSW onset according to the mean zonal wind at 10 hPa and 60°N. During this period, the zonal wind becomes westward, with wind reversal anomalies reaching approximately 8 m/s at 80 km altitude.

As we move southward from Svalbard to Tromsø and Sodankylä, the westward winds gradually strengthen after day 0, peaking at around 90 km with anomalies reaching up to 20 m/s. Following the onset of the SSW, eastward winds dominate from 80 to 85 km for approximately 45 days, coincidental with the presence of an elevated stratopause that forms after the stratospheric warming and the reformation of the 'normal' stratopause towards the spring transition (Manney et al., 2009; Matthias et al., 2021). In contrast, the meridional winds at these polar-latitude stations exhibit greater variability than the zonal winds. Alternating positive and negative wind speed anomalies are observed throughout the entire altitude range, both before and after the SSW onset (from day -50 to day 20), indicating significant large-scale planetary wave activity in the meridional winds before and during the SSW onset and a reduced activity afterward. The polar vortex is reestablished after the SSW, as indicated here by the period of intensified westerly winds, where the planetary waves are rather weak. Planetary wave activity during SSWs plays a crucial role in modulating atmospheric tides (Hibbins et al., 2019; Zhang et al., 2021; van Caspel et al., 2023; Qiao et al., 2024).

### 3.3 Composite analysis of the mean response of tides during SSW events

The tidal components have been extracted from the zonal and meridional winds observed by meteor radars following the procedure for the mean winds, leveraging the ASF2D analysis during SSW events. Figure 6 shows the composite amplitude anomalies for 10 SSWs of the TDT in the zonal and meridional winds at the three high-latitude stations, while Figures 7 and 8 illustrate the corresponding anomalies for SDT and DT, respectively. The anomalies for SDT and DT observed from WACCM-X(SD) are shown in Appendix A.

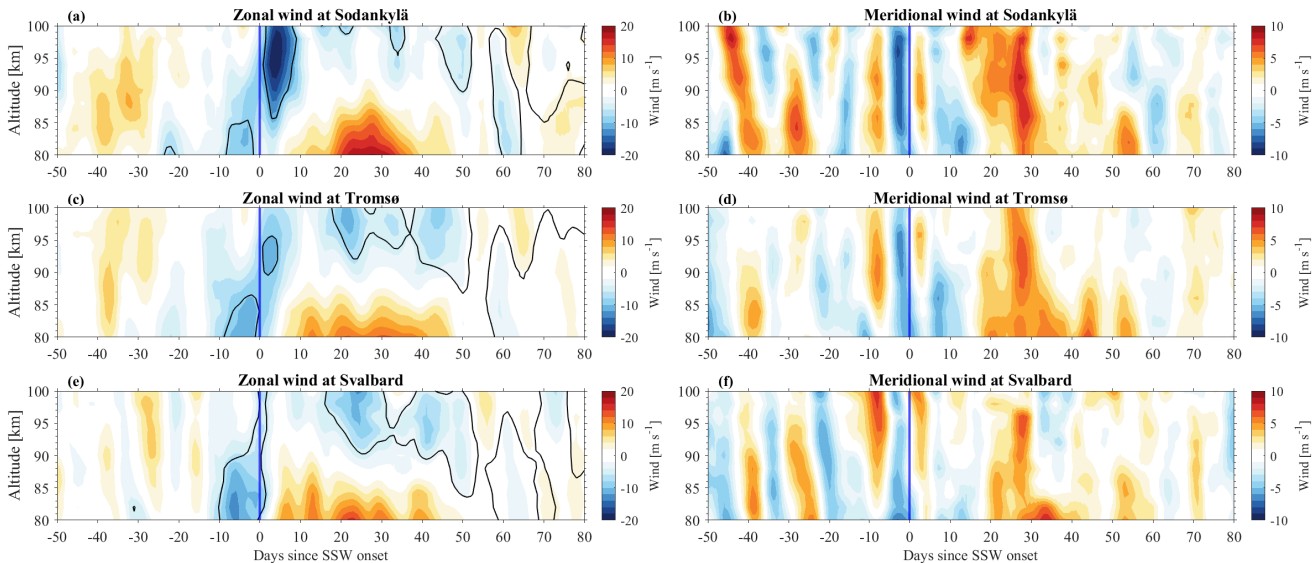

**Figure 5.** The cross-section of SSW composite (a, c, e) zonal and (b, d, f) meridional wind anomalies observed with three MRs at Sodankylä, Tromsø, and Svalbard over the period from 2004 to 2021, respectively. The vertical blue line represents the central date of the SSW (day 0 of SSW onset). The contours represent zero wind speeds.

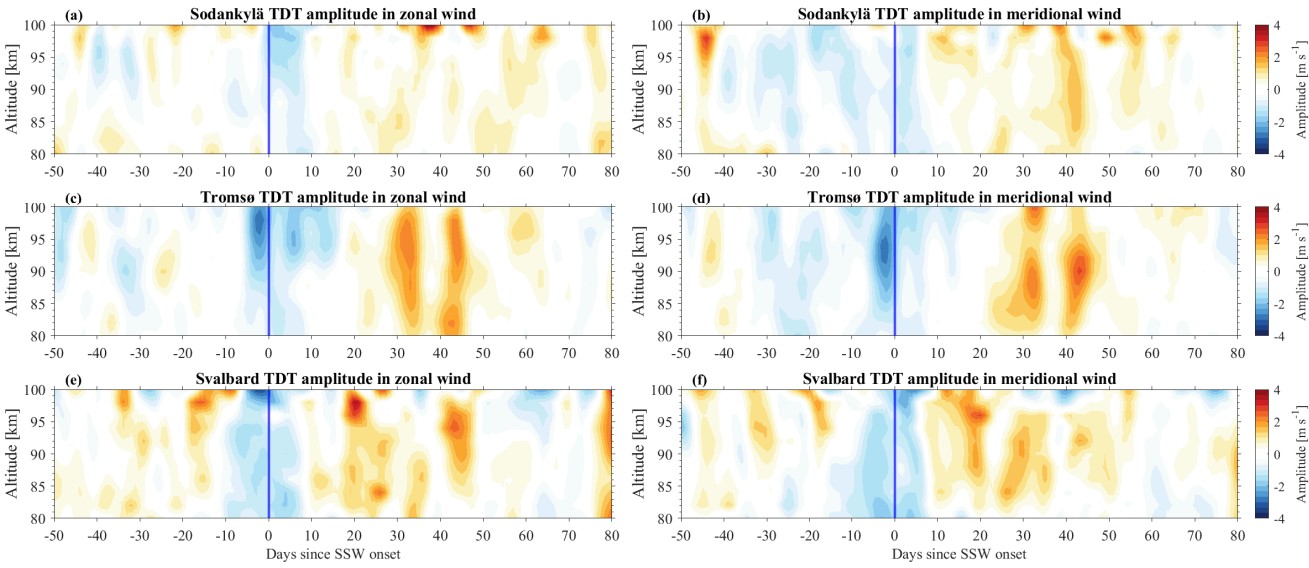

**Figure 6.** The cross-section of SSW composite TDT amplitude anomalies in the (a, c, e) zonal and (b, d, f) meridional wind components observed with three MRs at Sodankylä, Tromsø, and Svalbard, respectively.

In Figure 6, a pronounced decrease in TDT amplitude is observed within a few days before and after the SSW onset, characterized by a magnitude of up to -3 m/s in both the zonal and meridional wind components. The tidal amplitude changes can be presented as a percentage difference relative to the mean value. This reduction in TDT amplitude (20-30%) suggests a significant suppression of the TDT during the onset phase of SSWs, possibly due to changes in the background wind and temperature conditions that inhibit the propagation of this tidal component. Following this initial suppression, TDT amplitudes start to strengthen approximately 10 days after the SSW onset at Svalbard and Sodankylä and 20 days later at Tromsø, eventually reaching maximum positive anomalies of up to 4 m/s. The recovery and subsequent amplification of the TDT may be attributed to the reformation of the stratopause and the altered wind structure in the MLT region, which facilitates the upward propagation of the tides. Notably, the strength of the TDT amplitude anomaly at Tromsø and Svalbard is stronger than at Sodankylä, indicating potential latitudinal and longitudinal variations in the tidal response to SSWs. These observations align with previous studies that have utilized mechanistic global circulation models (Lilienthal et al., 2018; Lilienthal and Jacobi, 2019) and satellite measurements (Moudden and Forbes, 2013) to discuss the excitation mechanisms of TDT, such as direct solar heating, nonlinear interactions, and gravity wave–tide interactions.

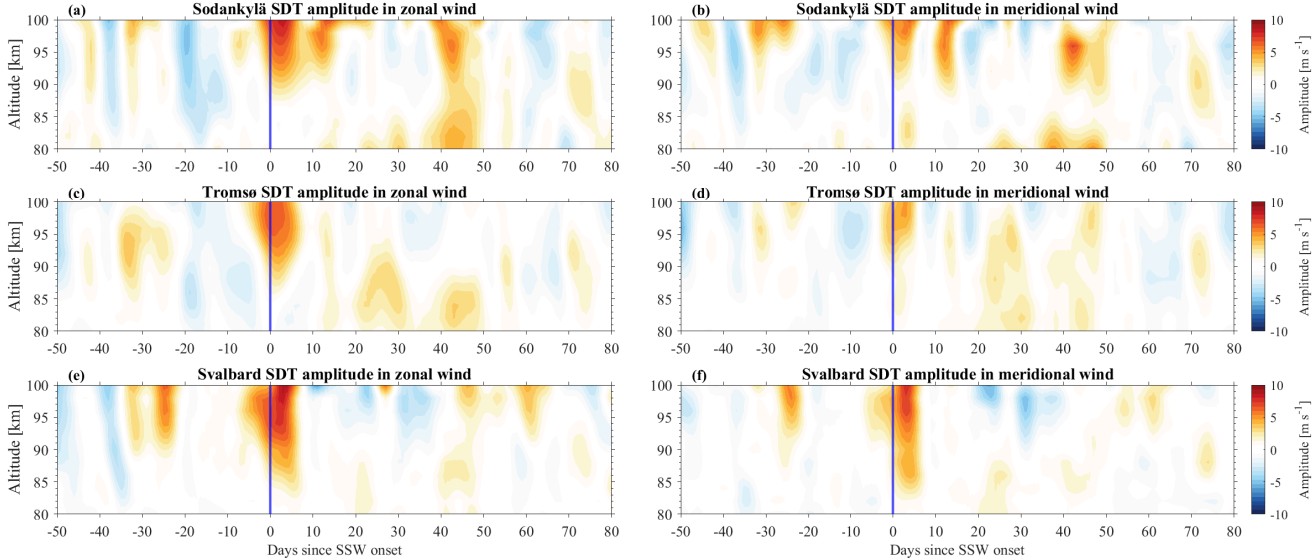

**Figure 7.** The same as Figure. 6 but for SDT amplitude anomalies.

Figure 7 illustrates that the SDT amplitude anomalies at the three stations, in both wind components, can reach up to 10 m/s (changes with peaks reaching up to 40%) within an altitude range of 90-100 km, persisting for only a few days around the SSW onset. This short-lived but intense enhancement of SDT, particularly in the zonal wind component, indicates a strong and rapid response of the SDT to the dynamical changes induced by the SSW. The pronounced increase in SDT amplitudes suggests that the modification of zonal mean winds during SSW may provide more favorable conditions for the upward propagation of SDT, potentially amplified by interactions with planetary waves. This rapid response is also consistent with the ozone enhancement

at equatorial to middle latitudes around the onset of the SSW, as shown by Siddiqui et al. (2019) (in Figure 3b). Overall, the SDT amplitude anomalies are found to be approximately twice as large as those of TDT and DT, highlighting the sensitivity of SDT to SSW-induced disturbances.

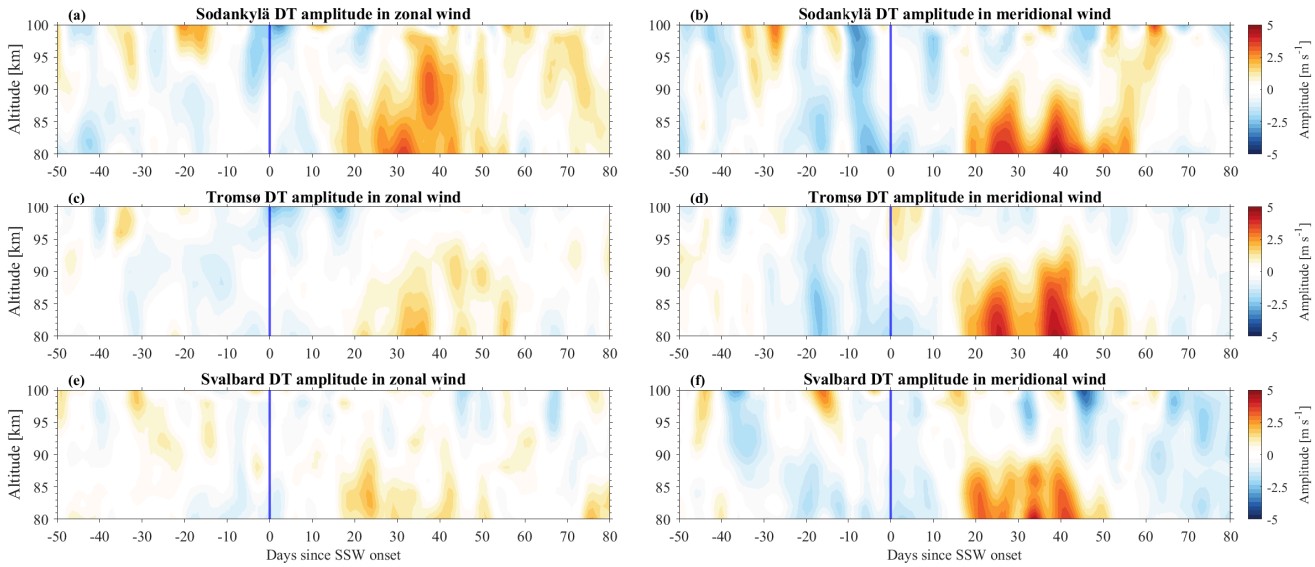

**Figure 8.** The same as Figure. 6 but for DT amplitude anomalies.

Figure 8 shows the DT amplitudes before, during, and after the central date of SSWs. Unlike the TDT and SDT, the DT amplitudes do not exhibit a distinct variation around the SSW onset. However, in both wind components at Sodankylä, Tromsø, and Svalbard, the DT amplitude starts to enhance around 20 days after the SSW onset, reaching a peak nearly 40 days post-SSW. This enhancement in DT amplitude persists for about a month, indicating a delayed response to the SSW. The stronger positive amplitude anomalies are more evident in the meridional wind component than in the zonal wind component. In comparison, the

positive amplitude anomaly in the zonal wind component above Tromsø and Svalbard is less pronounced than at Sodankylä, suggesting a latitudinal/longitudinal dependency in the DT response. The DT amplitude changes have magnitudes of the order of 30% in the zonal component and 50% in the meridional component. The delayed enhancement of DT could be linked to the restoration of the stratopause by the gradual descent of an elevated stratopause accompanied by increased ozone VMR and subsequent changes in thermal and dynamical conditions in the upper mesosphere, which may alter the tidal propagation

environment and lead to the amplification of the DT.

## 3.4   Atmospheric composition and radiative processes during SSWs

Trace gases play a crucial role in the energetic and radiative balance in WACCM-X(SD). The modeled long-wave cooling and short-wave heating rates provided by the simulation require a good representation of the trace gas VMR. Therefore, we

extracted the WACCM-X(SD) water vapor and ozone fields for all three MR locations and used the retrieved ozone and water
vapor VMR from our radiometers MIAWARA-C and GROMOS-C in Ny-Ålesund, Svalbard.

### 3.4.1 Water vapor

Figure 9 compares MLS and WACCM-X(SD) water vapor VMR anomalies for the entire middle atmosphere. The composite
analysis exhibits a similar pattern between the satellite and the model. There is a weak tendency in WACCM-X(SD) to under-
estimate water vapor anomalies after the SSW event. During the SSW around the central day, larger deviations are only visible
above 75 km.

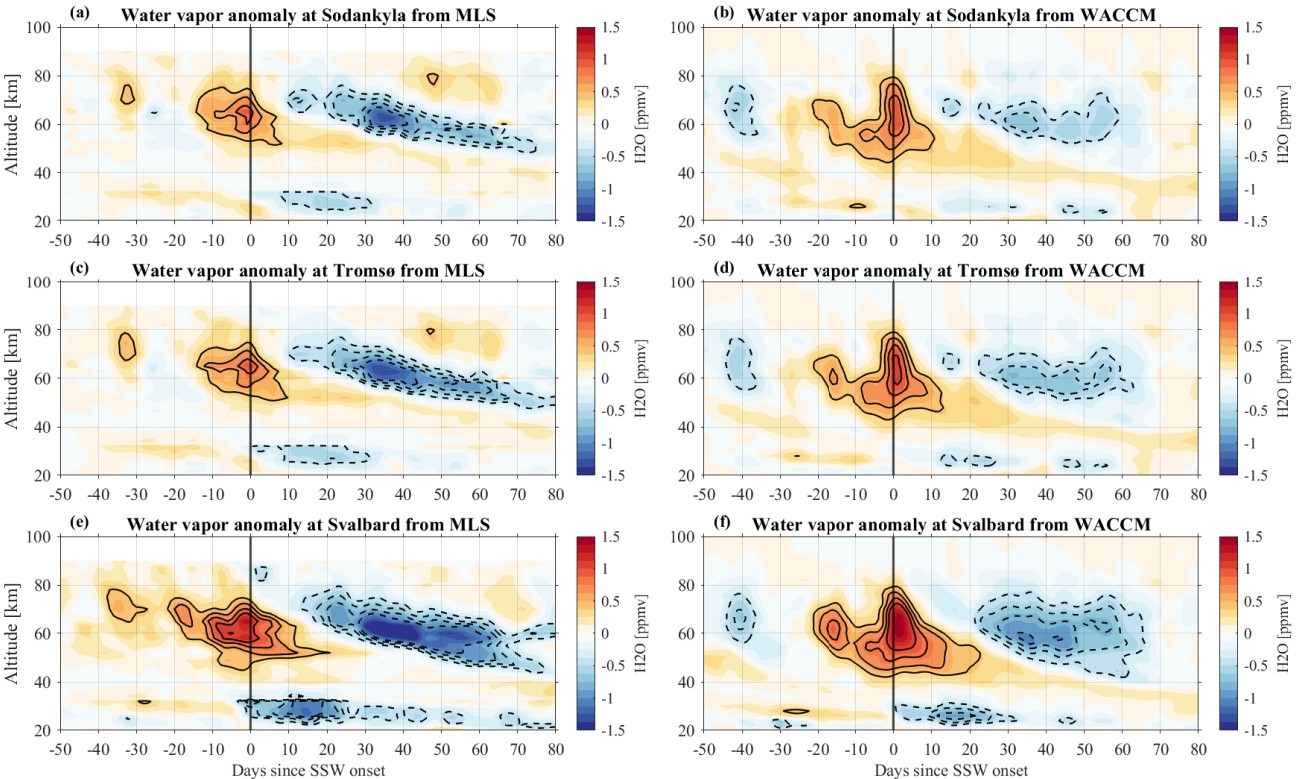

**Figure 9.** The cross-section of SSW composite water vapor anomalies measured by MLS (a, c, e) and simulated by WACCM-X(SD) (b,
d, f) over Sodankylä, Tromsø, and Svalbard. The positive and negative anomalies of water vapor are shown in solid and dashed contours,
respectively.

Furthermore, we compared the WACCM-X(SD) reduced water vapor VMRs with the local MIAWARA-C observations
from Ny-Ålesund. MIAWARA-C collects spectra during an entire day, whereas MLS only measures at fixed local times above
Svalbard. Figure 10 shows MIAWARA-C anomalies from the composite analysis between 2015-2023. The WACCM-X(SD)
contours are overlaid and reflect the good agreement for the water vapor anomalies. Composite analysis results in water vapor

VMR changes of about $\pm 1.25$ ppmv for SSW events. During the events, the intrusion of water vapor-rich airmasses from the mid-latitudes results in increased water vapor mixing ratios throughout the upper stratosphere up to the mesosphere, reaching altitudes of about 80 km. WACCM-X(SD) reproduces the SSW-related changes in water vapor VMR at the upper stratosphere and mesosphere. This is remarkable considering that the water vapor fields are not nudged and calculated from the interactive chemistry module. This also provides confidence that the calculated QRL cooling rate provides a meaningful model parameter for comparison to the observations.

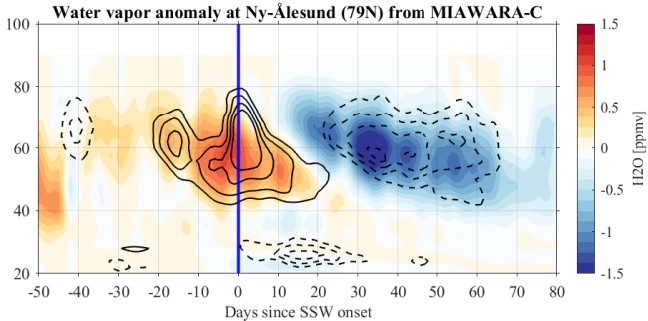

**Figure 10.** The cross-section of SSW composite water vapor anomalies (color-coded) from ground-based MIAWARA-C microwave radiometer measurements at Ny-Ålesund, Svalbard. The overlaid solid and dashed contours represent positive and negative anomalies of water vapor from WACCM-X(SD) at Svalbard, respectively.

### 3.4.2 Ozone

Ozone is one of the most crucial trace gases for the excitation of tides. WACCM-X(SD) shows all three ozone layers. The primary layer extends from 20-50 km, the tertiary ozone layer is found at altitudes between 60-80 km, and the secondary ozone layer covers altitudes above 90 km in Figure A2. At polar and high latitudes, WACCM-X(SD) exhibits the highest ozone VMR
at the secondary ozone layer, reaching 6-8 ppmv. The tertiary ozone layer takes values between 1-2 ppmv, and the stratospheric ozone reaches about 4 ppmv before the SSW and 5-6 ppmv in the aftermath. Figure 11 presents the ozone anomalies from MLS and WACCM-X(SD) corresponding to three stations during SSWs. The most significant positive ozone anomalies are up to 1.5 ppmv and persist for over 30 days in the middle stratosphere following the SSW onset. In the upper stratosphere and lower mesosphere, positive ozone anomalies persist for approximately two months starting around 10 days after the onset.
The persistence of positive ozone anomalies reflects prolonged changes in middle atmospheric circulation following the SSW. The most significant changes in ozone VMR occur in the secondary ozone layer and the stratosphere. The stratospheric ozone anomaly is closely related to the elevated stratopause and lasts up to 70 days after the SSW event. The secondary ozone layer indicates a response around the SSW event, with anomalies reaching a maximum 10-15 days after the SSW. The observed ozone increase reaches up to 1.5 ppmv, which is about 50% of the background value.

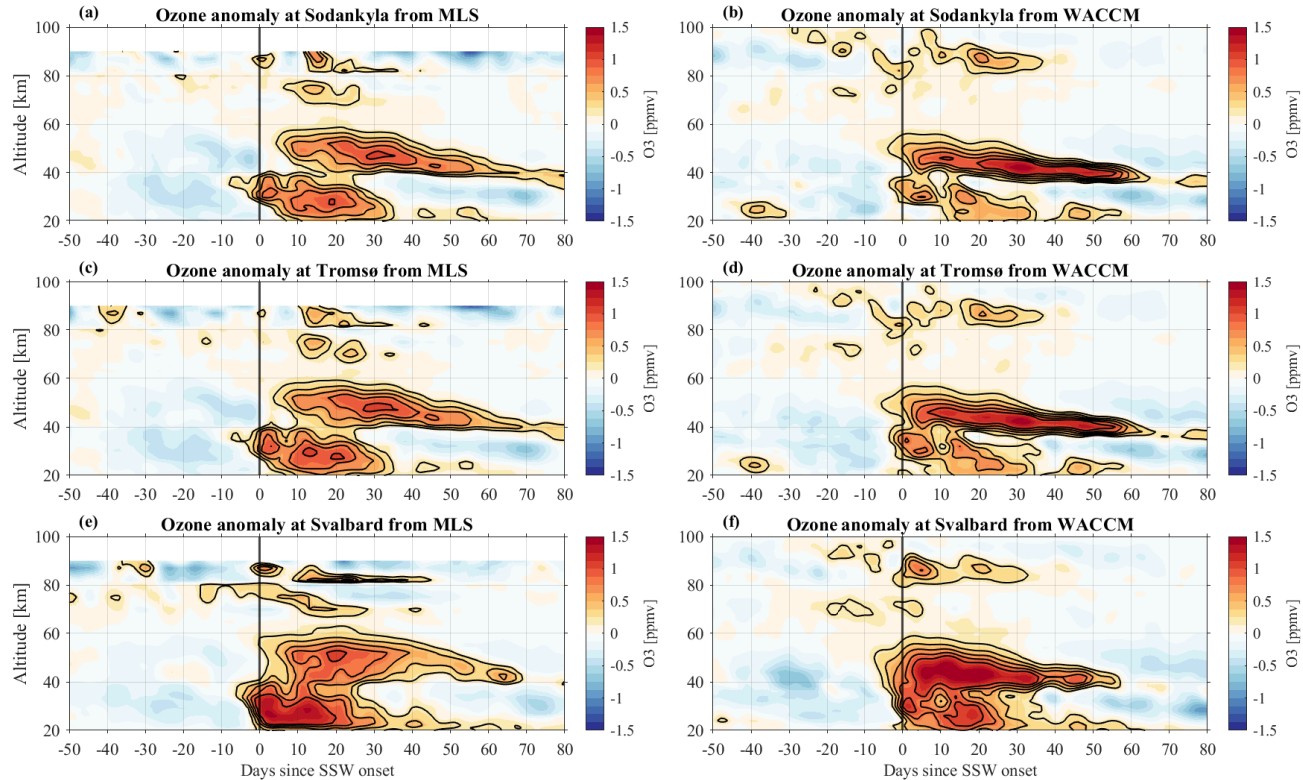

**Figure 11.** The cross-section of SSW composite ozone anomalies measured by MLS (a, c, e) and simulated by WACCM-X(SD) (b, d, f) over Sodankylä, Tromsø, and Svalbard. The positive anomalies of ozone are shown in solid contours.

We also performed a comparison between WACCM-X(SD), leveraging data reduction, and our local GROMOS-C measurements at Ny-Ålesund, Svalbard. Figure 12 shows in color scale the ozone VMR anomalies from the radiometer, and the WACCM-X(SD) contours are overlaid. The stratospheric ozone anomalies are well-reproduced in WACCM-X(SD), underlining once more the performance of the interactive chemistry module. The differences at altitudes above 75 km are mostly due to the low measurement response of GROMOS-C, as the spectral resolution is no longer sufficient to resolve the ozone Doppler broadened line peak at 142 GHz, thus, the retrieved ozone values start to become dominated by the a priori information taken from a climatology. Specifically, the ozone double-layer structure in the WACCM-X(SD) ozone anomalies that form at the onset of the SSW and last for about two weeks, results in two layers of substantial UV heating. The superposition of these two diurnal tidal waves at the mesosphere may effectively amplify the SDT at high latitudes due to a 12-hour phase offset caused by the different vertical distances both waves have to travel, considering the typical vertical wavelengths of 30-50 km for semidiurnal tides at this latitude (Stober et al., 2021b, 2020). The change in ozone is strongly dependent on altitude and latitude, as shown in the measurements and simulations (Figure 11). With the polar wind reversal during the SSW onset, planetary wave activity in the stratosphere drives anomalous equatorial upwelling and cooling that enhances tropical stratospheric ozone. This

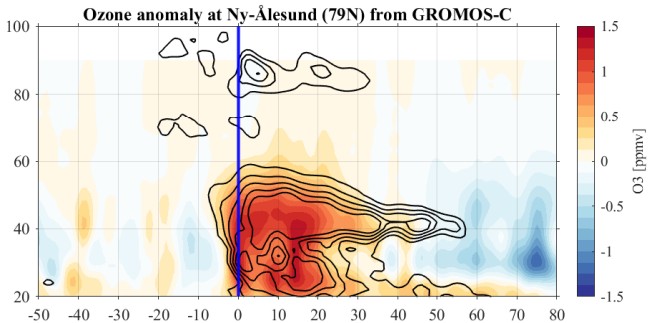

**Figure 12.** The cross-section of SSW composite water vapor anomalies from ground-based microwave radiometers GROMOS-C at Ny-Ålesund, Svalbard. The overlaid solid contours represent positive anomalies of ozone from WACCM-X(SD) at Svalbard.

is qualitatively consistent with findings from Siddiqui et al. (2019) and Limpasuvan et al. (2016), which show a rapid ozone increase from 20°S to 40°N, and a decrease poleward of 40°N.

### 3.4.3 QRL cooling and QRS heating rates

Previous studies have identified multiple mechanisms that could contribute to tidal variations during SSW events:

(a) Changes in the zonal mean winds that influence the vertical propagation of tides in the MLT region (Jin et al., 2012; Pedatella et al., 2012; Stober et al., 2020).

(b) Ozone changes (Goncharenko et al., 2012; Eswaraiah et al., 2018; Siddiqui et al., 2019; Mitra et al., 2024).

(c) Nonlinear interactions between stationary planetary waves and tides (Pedatella and Forbes, 2010; Pedatella and Liu, 2013; Qiao et al., 2024).

Performing the composite analysis, we derived QRL and QRS anomalies for all SSW events. These anomalies are shown in Figure 13 for Sodankylä, Tromsø, and Svalbard. There is almost no dependency on latitude in the general morphology of QRL cooling rate anomalies among the three stations. The elevated stratopause reflects reduced cooling rates (reddish color) and increased cooling above. The biggest anomalies and abrupt changes occur directly around and after the central day of the SSW, corresponding to the temperature changes (Figure A1). The QRS heating rate (including effects by ozone) exhibits a more clear stratification. The MLT region (80–100 km) exhibits pronounced positive heating rate anomalies around and shortly after the central date of the SSW, followed by a period of diminished, yet still positive, heating anomalies persisting from approximately 20 to 40 days after the event. Furthermore, the QRS shows a good correspondence to the ozone VMR and captures the diurnal cycle in the stratosphere. During the recovery phase of the stratopause, the QRS shows a characteristic diurnal cycle.

Given the close agreement between all 3 stations, we are going to focus on Tromsø when investigating the absolute QRL and QRS shown in Figure 14. The QRL absolute cooling rate indicates a rather consistent behavior before the SSW, with only a small cooling rate below 40 km and around 80 km altitude. Between these layers, we find cooling rates ranging from -10 to

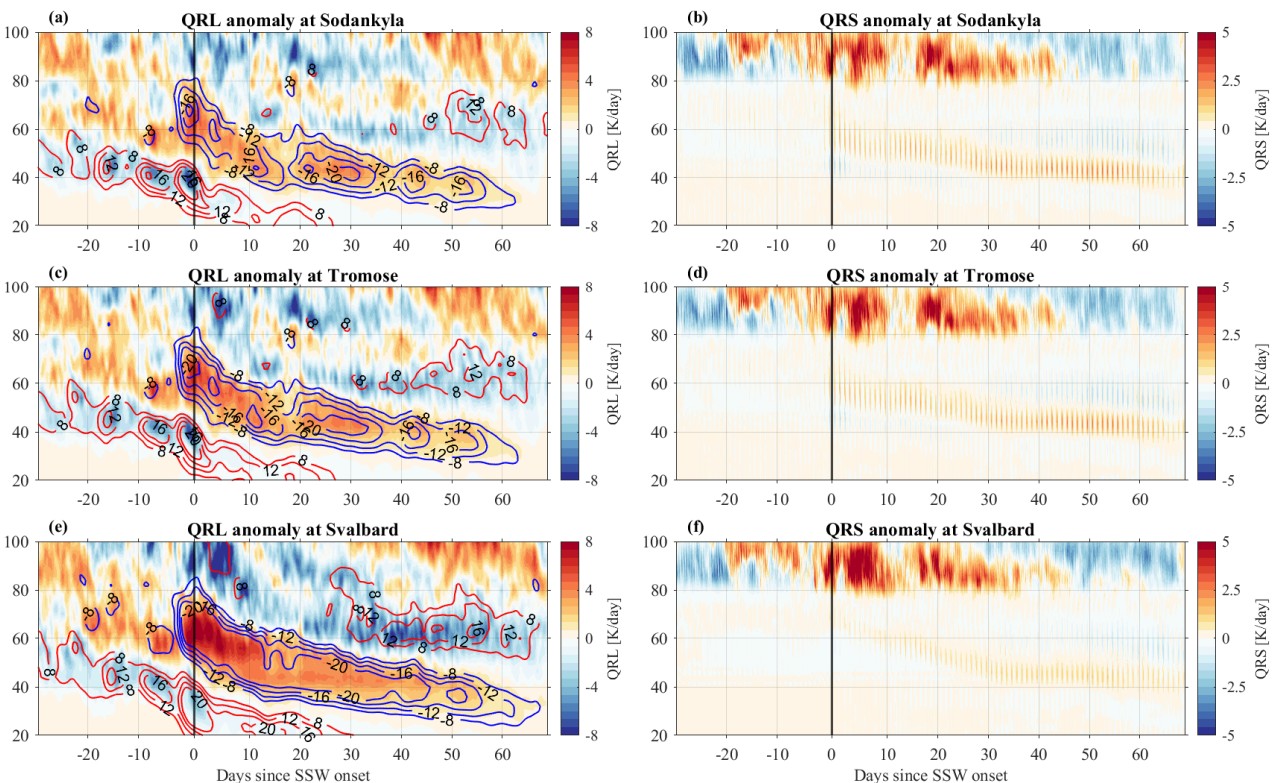

**Figure 13.** The cross-section of SSW composite QRL cooling (a, c, e) and QRS heating (b, d, f) rate anomalies simulated by WACCM-X(SD) over Sodankylä, Tromsø, and Svalbard. Red contours indicate positive temperature anomalies. Blue contours indicate negative temperature anomalies.

-15 K/day. During the SSW and the first 10-15 days after the SSW event, we find reduced cooling rates from the stratosphere up to 80 km in the MLT. The most extreme QRL value is observed after the central day above 80 km altitude, exceeding the values of -20 k/day. On the other side, the QRS heating aligns with the total ozone VMR at the secondary and primary ozone layers, and a much lower level also in the tertiary ozone layer. The largest QRS values are observed after the central day of the SSW at the MLT, and subsequently in the stratosphere about 20 to 60 days later, when the elevated stratopause reaches again the typical stratospheric altitudes. The QRS values at the MLT and the stratosphere are modulated by a diurnal cycle. For tidal excitation, the QRS plays the most important role in understanding the derived tidal anomalies from the MR. The QRL shows only a large-scale response with characteristic time scales of days during the SSw event. The cooling rates define the large-scale temperature and wind background through which the atmospheric tides have to propagate rather than contribute to the tidal forcing itself. Therefore, we conducted the composite analysis of the water vapor and ozone VMR, as shown in Figure 15. The water vapor VMR shows a characteristic response after the SSW and indicates the presence of an elevated stratopause, and towards the spring, a transition to the summer situation. However, the composite analysis of ozone VMR

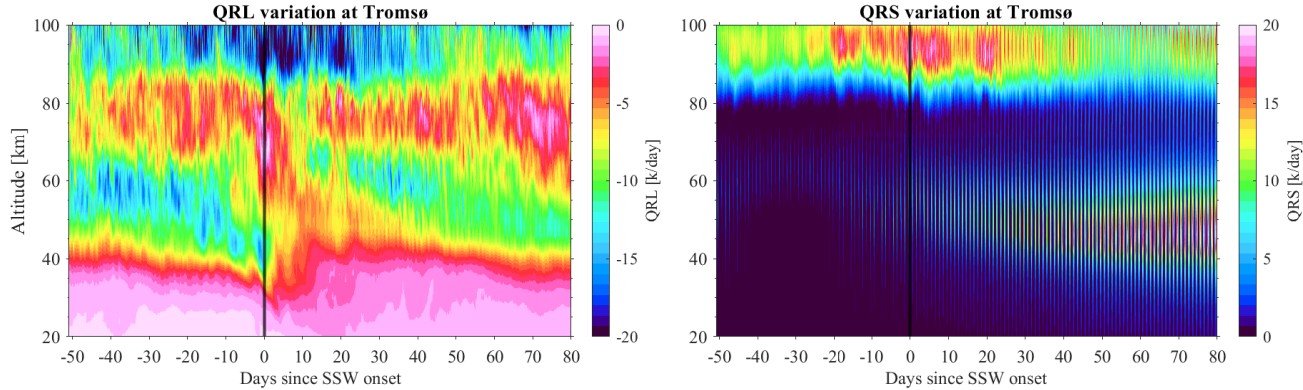

**Figure 14.** The cross-section of SSW composite QRL and QRS from WACCM-X(SD) at Tromsø.

indicates characteristic diurnal modulations at the stratosphere (30-50km) and at the altitude of the secondary ozone layer. This double-layer structure is mainly responsible for the semidiurnal tidal activity during the winter months at these latitudes. The superposition of the in-situ forced tide at the secondary ozone layer is out of phase with the diurnal tide that was excited at the stratosphere/lower mesosphere and propagates upward. The tidal phase lines shown for the 2018/19 case study reach the MLT region with a 12-hour offset compared to the diurnal heating cycle (see WACCM-X(SD) Figure 1). This coincides with the vertical wavelengths that are found in climatological analysis (Stober et al., 2021b) for these stations. During the SSW, this balance is disturbed, and the secondary ozone layer, together with the elevated stratopause, provides favorable conditions to amplify the SDT. The QRS anomaly shows a pronounced enhancement at the secondary ozone layer that coincides with the time of the semidiurnal tidal amplification and an increased ozone VMR at the stratosphere right after the SSW event. However, WACCM-X(SD) also exhibits a secondary peak of the semidiurnal tides about 20 days before the SSW (Figure A2), which is mostly absent in the observations. Apparently, with the onset of the SSW event, the dynamical conditions in the stratosphere and mesosphere, together with a sufficient ozone VMR at all three ozone layers, provide favorable conditions to cause the semidiurnal tide amplification. Furthermore, WACCM-X(SD) exhibits a fading of the QRS heating 20 days after the SSW event, which is accompanied by reduced semidiurnal tidal activity. During this period, the diurnal tide starts to dominate the observations around 80 km. This diurnal tide seems to be the direct result of the diurnal ozone cycle, which is associated with QRS in the stratosphere. The delayed response of QRS (and consequently DT) in the stratosphere is likely associated with the seasonal increase in solar illumination, as sunlight progressively reaches lower altitudes.

A crucial aspect of marked changes in direct shortwave heating of the stratosphere results from the increased ozone VMR. Therefore, we performed additional analysis by computing the circle of illumination in dependence on altitude. Figure 16 shows a schematic of how to estimate the presence of sunlight for a specific geographic latitude and local time. All calculations are performed using the J2000 reference epoch, and the sun ephemerides are valid for the period from 1950 to 2050 (United States Naval Observatory. Nautical Almanac Office and Nautical Almanac Office (U.S.), 2009). The sun is above the horizon at Svalbard at local noon after the 15th of January in the mesosphere and reaches the stratospheric altitudes towards the end

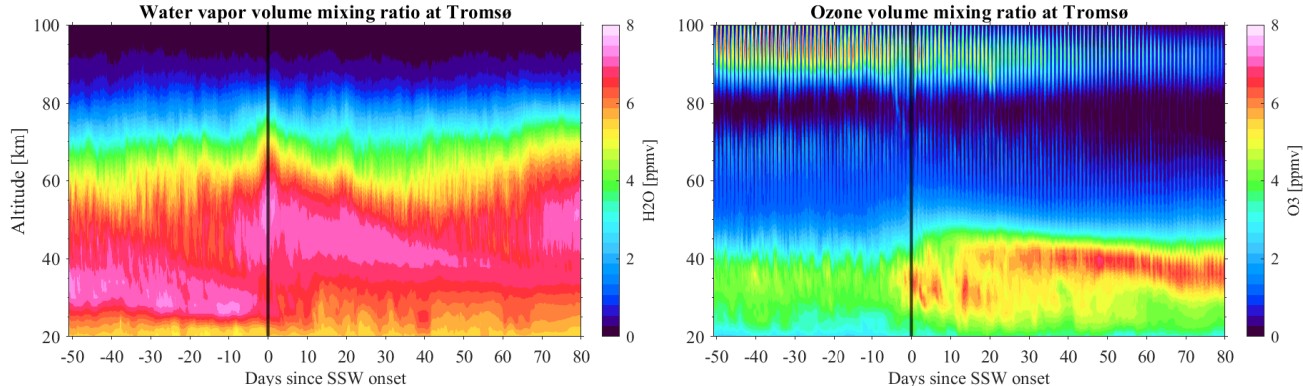

**Figure 15.** The cross-section of SSW composite of water vapor (left) and ozone (right) VMR anomalies from WACCM-X(SD) at Tromsø.

of January. At Tromsø and Sodankylä, the stratosphere is always illuminated by the sun during noon, although both sites are
at latitudes beyond the polar circle and the surface remains in darkness. During midnight, all altitudes remain shadowed by
the Earth, and no sunlight reaches the stratosphere or mesosphere. This characteristic diurnal heating results in a pronounced
diurnal tide at the stratosphere (Schranz et al., 2018). Thus, the differences in the QRS between Svalbard and the Fennoscan-
dinavian mainland locations are understandable due to the availability of direct sunlight at certain altitudes. We calculated the

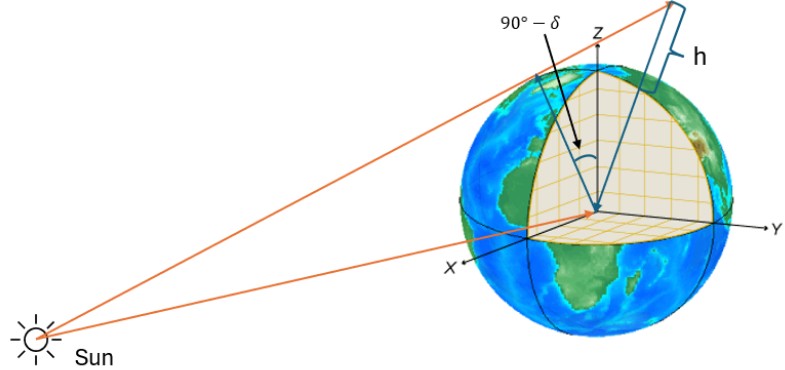

**Figure 16.** A schematic illustration showing the estimated sunlight presence for a given geographic latitude and local time. The diagram
visualizes the variation in sunlight exposure as a function of latitude, highlighting the effects of Earth's rotation and axial tilt on day and
night cycles.

circle of illumination height above the latitudes of Sodankylä, Tromsø, and Svalbard for the longitude of the Greenwich merid-
ian, assuming 11:59:59 UTC for the dates listed in Table 1. These numbers are valid for 2023 and depend on the exact solar
longitude of the Earth around the sun's orbit. Due to leap years, there are some changes in the exact altitudes between years of
a few hundred meters. We show the lowest altitude, the sun will be visible for that latitude at the Greenwich meridian for that

**Table 1.** Illumination height at 12 UT above the latitudes of Sodankylä, Tromsø, and Svalbard.

| Date | Sodankylä (67.37°N) | Tromsø (69.58°N) | Svalbard(78.99°N) |
|------|---------------------|------------------|-------------------|
| 01 Jan | 0.14km | 4 | 113km |
| 15 Jan | 0 | 0.23km | 77km |
| 31 Jan | 0 | 0 | 26km |
| 15 Feb | 0 | 0 | 0 |
| 28 Feb | 0 | 0 | 0 |

date, around noon (UTC). A zero indicates that the sun reaches the surface. Furthermore, considering the effective sampling volume of more than 400 km in diameter of GROMOS-C and the meteor radars, some parts of the observation volumes are further south and, thus, can collect sunlight at even lower altitudes.

### 3.5 Diurnal tide amplification and radiative forcing

The amplification of diurnal tides observed 20–40 days after the SSW is closely linked to enhanced radiative forcing, which is closely related to the increased ozone VMR due to the formation of an elevated stratopause, combined with increasing solar radiation toward the end of January. During this period, WACCM-X(SD) exhibits a pronounced diurnal amplitude enhancement around 60-80 km. MR observations reveal a similar amplification, extending 5–10 km higher into the mesosphere. Both model and observations indicate that the meridional wind component shows a stronger enhancement than the zonal component, as shown in Figure 17, which compares MR and WACCM-X(SD) DW1 wind perturbations.

The wind perturbations show clear evidence of vertically propagation tidal modes, as indicated by the vertical phase progression. Notably, DW1 amplitudes in WACCM-X(SD) diminish between 90 and 100 km altitude, precisely the region where the SW2 tide begins to grow significantly. MR observations also display a gradual decrease in DW1 amplitude with height, though the trend is less pronounced than in the model. The DW1 temperature perturbations in WACCM-X(SD) are in good agreement with the meridional wind structure and remain in phase with the QRS heating near 40 km altitude, corresponding to the warm phase of the tide driven by solar forcing. We overlaid the QRS perturbation at the stratosphere onto the wind and temperature contours. Figure 17e compares the QRS and temperature perturbation for the DW1 frequency component averaged between 34-44 km altitude, corresponding to the seed region of the diurnal tidal wave (altitude region before the amplitude grows rapidly). A modest phase offset is apparent between both time series due to the spike-like profile of the QRS and temperature perturbation, in that QRS essentially drops to zero during the night and rises sharply during the day. QRS appears to lead the temperature response. This phase offset may also be influenced by the coarse 3-hour temporal resolution of the WACCM-X(SD) output.

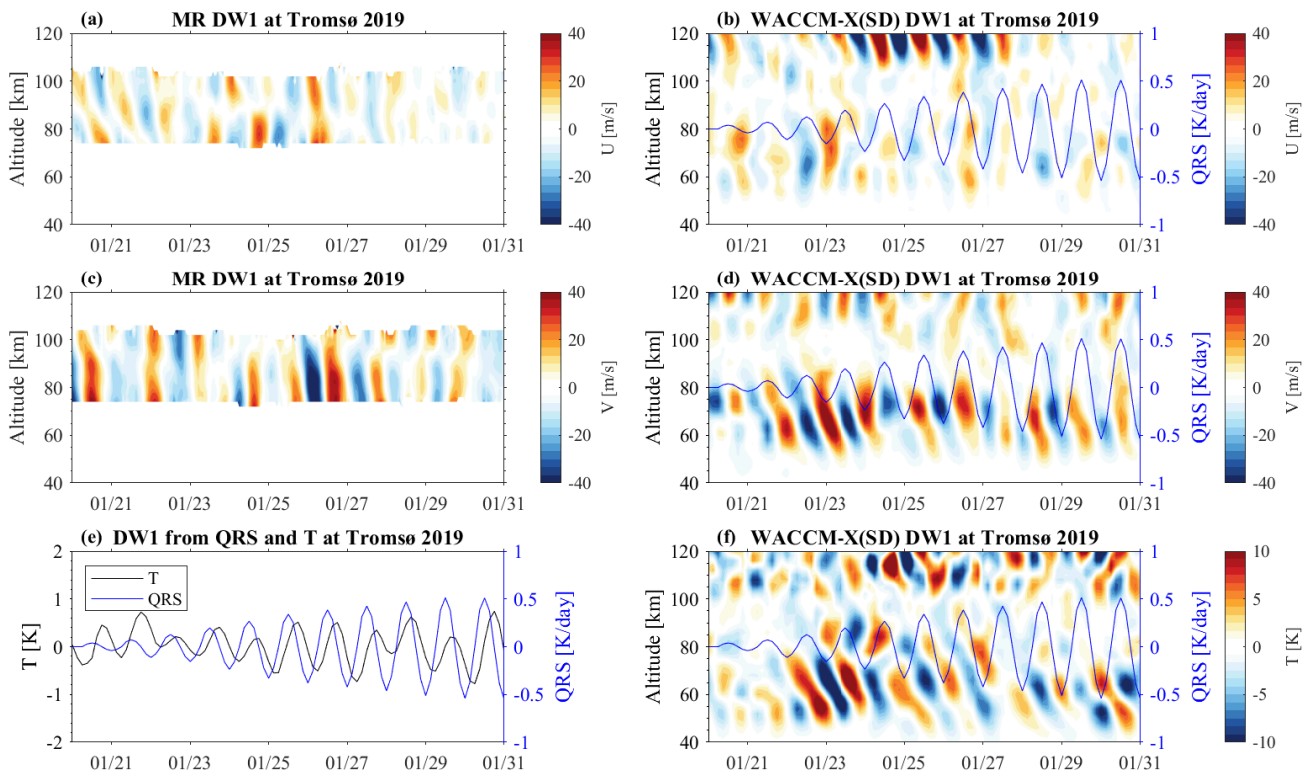

**Figure 17.** Time series of DW1 zonal and meridional winds, temperature, and QRS perturbations derived using adaptive spectral filtering over Tromsø during 20–40 days following the 2018/19 SSW event. Panels (a) and (c) show results from MR observations, while panels (b), (d), and (f) show results from WACCM-X(SD) simulations. The median values of temperature (black line) and QRS (blue line) perturbations in the stratosphere (34-44 km altitude range) are presented in (e).

## 3.6 Semidiurnal tide amplification during SSW

The amplification of the semidiurnal tide during SSW events was reported in various previous studies (He et al., 2024; Forbes and Zhang, 2012; He and Chau, 2019), which primarily focused on wind and temperature from observations to investigate this phenomenon. Here, we present a detailed case study comparison between MR observations and WACCM-X(SD) in wind and temperature fields to further investigate semidiurnal tidal behavior. Figure 18 shows the semidurnal tide in zonal and meridional wind perturbations from both MR and WACCM-X(SD), including the temperature fields. Furthermore, we extracted the SW2 spectral band from QRS at mesosphere averaged over the 84–92 km altitude and overlaid it on the WACCM-X(SD) results. Both model and observations exhibit a strong semidiurnal tide throughout most of the winter season above 90-100 km, with a clear vertical phase progression indicative of upward propagation. In MR observations, semidiurnal tide amplification becomes apparent after the onset of the SSW event, whereas in WACCM-X(SD), enhancement begins several days earlier. The simulated tidal amplitudes are also considerably larger than those observed by MR. Notably, both WACCM-X(SD) and MR data

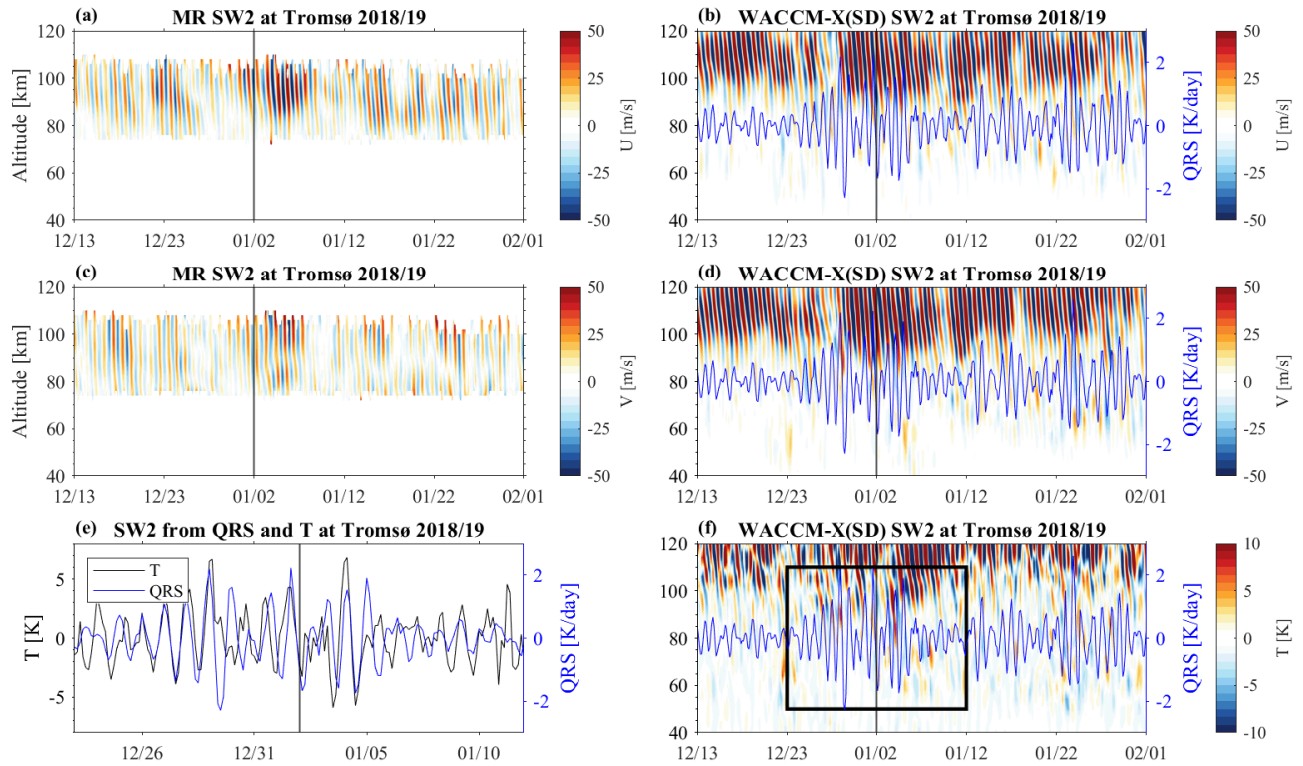

**Figure 18.** Time series of SW2 zonal and meridional winds, temperature, and QRS perturbations derived using adaptive spectral filtering over Tromsø before 20 days, and after 30 days following the SSW 2018/19 onset. Panels (a) and (c) show results from MR observations, while panels (b), (d), and (f) show results from WACCM-X(SD) simulations. A zoomed-out view of the time series (black solid lines) is shown in panel (f), corresponding to the median values of temperature (black line) and QRS (blue line) perturbations in the mesosphere (84-92 km altitude range) as presented in (e).

display distinct variability in the SW2 tide at multi-day to weekly timescales, which closely resembles modulations observed in the QRS. Filtering the QRS for the semidiurnal spectral contribution exhibits that several bursts of semidiurnal variation agree mostly in the time of their occurrence with an amplified semidiurnal tide in the winds in WACCM-X(SD). However, this coherence is much less visible in the semidiurnal temperature perturbations. Although our spectral analysis of the QRS reveals

a semidiurnal variability, the physical mechanism driving this modulation remains uncertain. Nonetheless, such a semidiurnal modulation is beneficial to amplify the semidiurnal tide higher up in the MLT during the SSW.

We further examine the relationship between semidiurnal perturbation in QRS and temperature in Figure 18e. In the tidal amplification region between 82–92 km, the semidiurnal perturbation in QRS reaches approximately 50% of the temperature perturbation and shows even good agreement with periods of enhanced temperature amplitudes. Above this layer, both semid-

440 iurnal tidal wind and temperature amplitudes grow rapidly in WACCM-X(SD). MR tidal amplitudes also exhibit increasing magnitude with increasing height, but their amplitude reaches approximately 50% of the WACCM-X(SD).

## 4    Discussion

This study investigates the amplitude anomalies of DT, SDT, and TDT tides during SSW events using MR data from three polar-latitude stations. The combined analysis of tidal amplitude anomalies and trace gas variations provides new insights into the factors influencing tidal dynamics during SSWs, with a specific focus on the roles of ozone and water vapor. Our analysis of MR data reveals distinct responses of DT, SDT, and TDT during SSW events. Combining observations and model simulation results represents the attempt to explain mesospheric tidal variability in polar regions during SSWs by highlighting the critical roles of trace gases, and short-wave and long-wave radiative heating.

The sun might not be above the horizon all the time during the day, and, hence, the effective short-wave absorption might be smaller, which reduces the magnitude of the diurnal temperature change. In addition, it is important to emphasize that the observed tidal change is part of the global tidal response, and the heating/cooling driving those changes are not confined to winter high latitudes. The observed changes are a proxy for the global change, as discussed in the next paragraph. However, this is not the case for the QRL, which means that the middle atmosphere is dominated by effective daily cooling rates of up to 10 to 20 K/day at the MLT. This cooling of the middle atmosphere defines the large-scale temperature field, and the stratification of the middle atmosphere between the polar and middle latitudes drives the circulation and provides the background condition for the propagation of planetary, tidal, and gravity waves. These dynamic effects further contribute to the energy balance and the vertical coupling between the layers and can reach similar magnitudes to the QRL (see Appendix C).

Furthermore, the WACCM-X(SD) QRL and QRS values result from various processes, with contributions from the radiative effects of ozone and water vapor (see also Liu et al. (2010b) and references therein). Our comparison to these trace gases, thus, provides a proxy for the radiative heating changes. The radiative forcing, summarized, e.g., in Andrews et al. (1987), underlines that ozone is the most important absorber of UV radiation at the stratosphere and lower mesosphere. However, in the MLT region, oxygen and other chemical trace gases, resulting in exothermic heating, become significant. This is important to consider at the MLT, where QRS often indicates a morphology very similar to the ozone VMR, but other chemical reactions and absorbing species already dominate the absolute value, although ozone might affect the reaction rates or VMR of these species as well. This is in particular the case for atomic oxygen.

The enhancement of the diurnal tide 20-40 days after the SSW provides important insights into post/SSW tidal dynamics. This amplification appears to result directly from the increased ozone VMR, associated with the formation of the elevated stratopause. WACCM-X(SD) shows a strong correspondence between the QRS-induced diurnal heating and the diurnal temperature tide at the stratosphere. Although this diurnal tide is supposed to be a trapped mode due to its negative equivalent depth, the WACCM-X(SD) model exhibits a vertical propagation in both temperature and wind perturbations. In the MLT regions (80–90 km), however, the upward-propagating diurnal tide appears to be annihilated and disappears. At these altitudes, the upward propagating diurnal tide is out of phase with the diurnal modulation of the QRS. Thus, heating occurs at heights corresponding to the secondary ozone layer. Given the energy and momentum conservation, the dissipated tidal energy may be transferred to other wave modes or the background mean flow. It is plausible that wave–wave interactions, such as nonlinear wave coupling, could facilitate the generation of secondary tides, potentially contributing to the observed semidiurnal tide

amplification. However, a detailed analysis of these processes is beyond the scope of this study.

Non-migrating tides are also affected by SSW events, though they exhibit considerable inter-event variability (Liu et al., 2010b; Miyoshi et al., 2017; Stober et al., 2020). Their forcing seems to depend on the longitudinal difference in the vertical propagation and longitudinal variations of the main absorbing trace gases and the planetary wave activity in the middle atmosphere.

In particular, ozone VMR is modulated by the planetary wave activity, which transports ozone-rich airmasses from the midlatitudes to high latitudes at stratospheric altitudes. Note, however, that we obtain the combined signature of migrating and non-migrating tides from single-station measurements.

We also analyzed observations from the southernmost MR available in the European sector. The Collm MR (51.3°N,13°E) is located at the edge of the polar vortex and, thus, sometimes shows signs of SSW events. As a result, the derived mean winds and

485 tidal anomalies show a morphology similar to that observed at polar and high latitudes. The Collm observations also reflect the enhancement of DT and SDT (see Appendix B1), suggesting a hemispheric impact of these anomalies from polar to the midlatitudes. A few degrees further south, observations from MIAWARA and GROMOS provide additional insights. Figure B2 shows composite water vapor and ozone VMR for the location at Zimmerwald/Bern (47°N, 7°E), Switzerland. Applying the same composite analysis to the ozone and water vapor measurements from GROMOS and MIAWARA at midlatitudes reveals

no discernible signatures associated with the SSW events. This is also reflected in the QRL and QRS anomalies shown in Appendix B3. The QRL and QRS cooling and heating rates indicate a more stratified vertical structure resembling the layering found in the trace gas measurements at the stratosphere and lower mesosphere.

One crucial aspect of the presented analysis is the implementation of the adaptive spectral filter technique, which is designed to infer the short-term tidal variability using an adaptive window length for each tidal mode, performing an onion peeling

scheme solving first for the mean wind and diurnal tide and other long-period oscillations, which are later used as background regularization for the higher frequency tidal components (Baumgarten and Stober, 2019; Stober et al., 2020). A critical aspect of the algorithm is that the number of wave cycles determines the window length for each tidal component that is extracted, which in the case of the TDT, SDT, and DT is very short, resulting in a rather wide bandwidth around each tidal frequency; thus, there could be some contamination due to gravity wave activity that falls into this band. The reduction of TDT amplitude during

the SSW might be a combined result of a weakening of the tidal generation and changes in GW activity in the stratosphere. A recent study has shown that the polar vortex itself can act as a source of GWs (Vadas et al., 2024), and this source diminishes during the SSW as the polar vortex breaks down.

Our results also confirm the results obtained in van Caspel et al. (2023) using the PRimitive equations In Sigma-coordinates Model (PRISM) tidal model and the Navy Global Environmental Model-High Altitude (NAVGEM-HA) background dynamics

to investigate the role of lunar tides relative to the SDT during SSWs. At Svalbard, the lunar tidal amplification should be minimal compared to the mid-latitudes. However, the presented analysis reveals a clear SDT amplification after the onset of the SSW, similar to Tromsø and Sodankylä. This points towards a minor contribution of the lunar tide, considering that lunar tidal potential changes with latitude (Vial and Forbes, 1994; van Caspel et al., 2023). Furthermore, lunar tides are also trapped tidal modes, resulting in a net zero vertical energy flux and requiring a similar mixing effect as proposed above to be amplified

(Perkeris resonance). Separating both mechanisms is crucial and requires spectral decomposition on a day-to-day basis. All

long window Fourier or wavelet-based methods require phase stability for the selected window length, which is about 21 days to separate a lunar tide from a semidiurnal tide. This assumption is not satisfied considering the day-to-day variability revealed by the adaptive spectral filter (Baumgarten and Stober, 2019; Stober et al., 2020). This underlines that radiative effects (heating/cooling) due to the increase/decrease of trace gases such as ozone play a key role in these short-term amplifications and can further amplify dynamical effects, altering the vertical tidal propagation during SSWs.

## 5 Conclusions

This study provides a comprehensive quantification of tidal variability and co-located tracer variability by combining observational data with model simulations. It discusses the radiative effects of tracer anomalies on mesospheric tidal variability during SSWs. Our analysis reveals distinct tidal responses to SSWs, characterized by a pronounced enhancement of SDT amplitudes following the onset of these events, while TDT amplitudes exhibit a decrease. The reduction in TDT during SSW likely results from a combination of diminished tidal generation and changes in gravity wave activity, which are strongly influenced by the breakdown of the polar vortex. The DT, in contrast, exhibits a delayed enhancement of amplitude, coinciding with the presence of an elevated stratopause and radiative heating effects driven by stratospheric ozone anomalies, which become prominent 20–30 days after the SSW onset. The observed SDT amplification is likely primarily due to the dynamical effect and modified by radiative heating effects modulated by ozone variability. Our correlation analysis between the semidiurnal tidal perturbation in QRS and temperature at the amplification region suggests that direct radiative absorption plays some role in the observed tidal enhancement. The ozone VMR exhibits distinct diurnal modulations at both the stratospheric ozone layer and the secondary ozone layer, forming a double-layer structure that plays a pivotal role in modulating semidiurnal tidal activity during winter at high latitudes, which leads to a constructive superposition of in situ and propagating DT components resulting in enhanced tidal SDT amplitudes in the MLT region. However, this proposed mechanism requires further investigation and confirmation in future studies.

Radiative processes, including both QRS heating and QRL cooling rates, play a fundamental role in shaping the large-scale temperature structure and atmospheric stratification, thereby modulating the background conditions for planetary, tidal, and gravity wave propagation. The longwave radiative effect dominates cooling processes in the MLT, with daily-averaged cooling rates reaching up to -20 K/day. The dominance of QRL at mesospheric and lower thermospheric altitudes underscores its significance in governing energy balance and vertical coupling processes in the middle atmosphere.

In summary, this study provides new insights into the coupling between trace gas variations, radiative heating, and tidal dynamics in the polar mesosphere during SSWs. Future work should further quantify the isolated radiative effects on tidal generation, potentially through targeted model simulations that distinguish between dynamical and radiative contributions.

*Code availability.* As a component of the community earth system model, WACCM-X source codes are publicly available at https://www. cesm.ucar.edu/ (NCAR, 2024).

*Data availability.* MLS v5 data are available from the NASA Goddard Space Flight Center Earth Sciences Data and Information Services Center (GES DISC): https://doi.org/10.5067/Aura/MLS/DATA2516 (Schwartz et al., 2020). The MR data can be obtained upon request from the instrument PIs. The 3-hourly WACCM-X simulation output is archived on NCAR's archive repository and can be obtained upon request from Guochun Shi. The GROMOS-C and MIAWARA-C level 2 data are provided by the Network for the Detection of Atmospheric Composition Change and are available at https://www-air.larc.nasa.gov/pub/NDACC/PUBLIC/meta/mwave/ (University of Bern, 2024). NOAA CSL: Chemistry & Climate Processes: SSWC, https://csl.noaa.gov/groups/csl8/sswcompendium/majorevents.html, (last access: October 2024).

*Author contributions.* GShi was responsible for the WACCM-X simulations, performed the data analysis, and prepared the manuscript. GStober contributed to the interpretation of the results. NG, MT, CJ, and AK provided MR data. HL and KW provided their technical assistance in setting up the WACCM-X simulations, and DP supported data reduction. All of the authors provided valuable feedback for manuscript editing.

*Competing interests.* The contact author has declared that none of the authors has any competing interests.

*Acknowledgements.* Guochun Shi and Gunter Stober are members of the Oeschger Center for Climate Change Research. We acknowledge the PIs for maintaining radar operations. We also thank Douglas Kinnison for helpful discussions and Joe McInerney for his technical assistance in setting up the WACCM-X simulations. Guochun Shi would like to acknowledge travel support provided by the High Altitude Observatory Visitor Committee. We thank the Aura/MLS team and NASA/JPL for providing the Microwave Limb Sounding measurements, with the level 2 data set available through the Aura Validation Data Center. Christoph Jacobi acknowledges the support of the IAP Kühlungsborn, Germany, for their support in the maintenance of the Collm radar. This research was supported by the International Space Science Institute (ISSI) in Bern, through ISSI International Team project # 23-580 "Meteors and phenomena at the boundary between Earth's atmosphere and outer space". National Center for Atmospheric Research is a major facility sponsored by the National Science Foundation under Cooperative Agreement No. 1852977.

*Financial support.* This research has been supported by the Swiss National Science Foundation (grant no. 200021-200517/1) and STFC UK grant (ST/W00089X/1). This research employed data from instruments supported by the Research Council of Norway under the project Svalbard Integrated Arctic Earth Observing System—Infrastructure Development of the Norwegian node (SIOS-InfraNor, Project No. 269927). The operation of MIAWARA-C and GROMOS-C is supported by AWIPEV under grant AWIPEV_0023.

## Appendix A: Tidal amplitudes from WACCM-X(SD)

## Appendix B: Mean winds and tidal amplitudes at mid-latitude station

**Table A1.** Dates of major SSW events were used for the composite in this study.

| Number | Winters | SSW central date |
| --- | --- | --- |
| 1 | 2005/2006 | 21 Jan 2006 |
| 2 | 2006/2007 | 24 Feb 2007 |
| 3 | 2007/2008 | 22 Feb 2008 |
| 4 | 2008/2009 | 24 Jan 2009 |
| 5 | 2009/2010 | 09 Feb 2010 |
| 6 | 2012/2013 | 06 Jan 2013 |
| 7 | 2017/2018 | 12 Feb 2018 |
| 8 | 2018/2019 | 02 Jan 2019 |
| 9 | 2020/2021 | 03 Jan 2021 |
| 10 | 2022/2023 | 16 Feb 2023 |

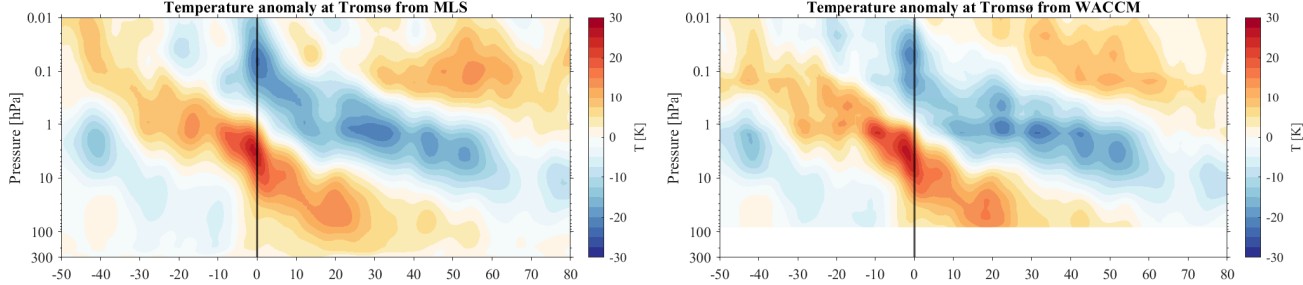

**Figure A1.** The cross-section of SSW composite temperature anomalies at Tromsø from MLS and WACCM-X(SD).

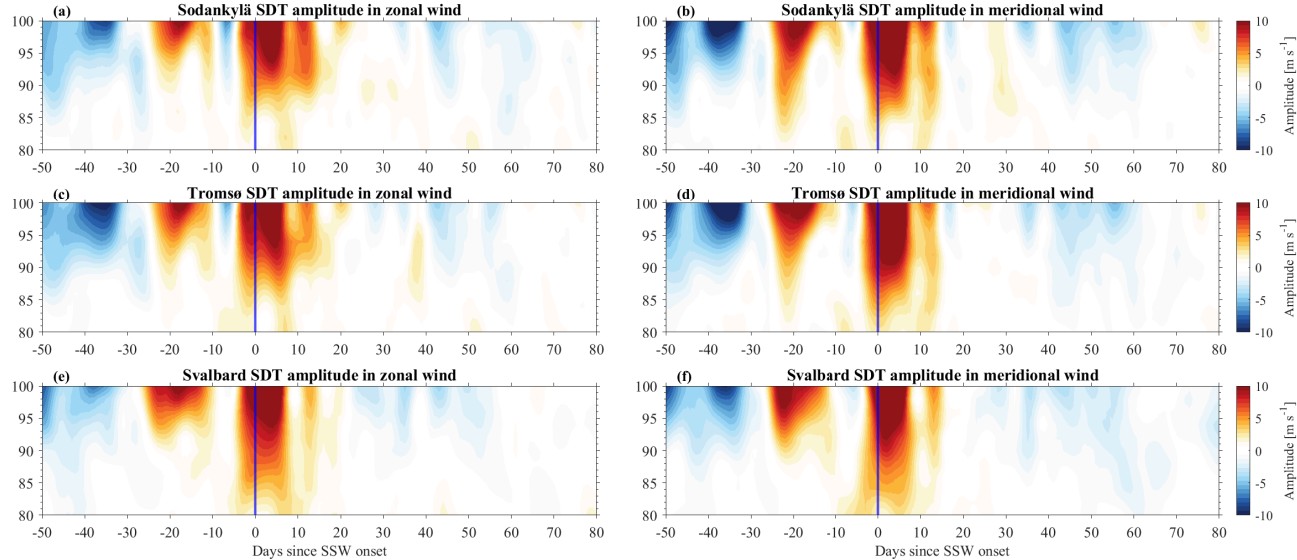

**Figure A2.** The same as Figure. 7 but for WACCM-X(SD).

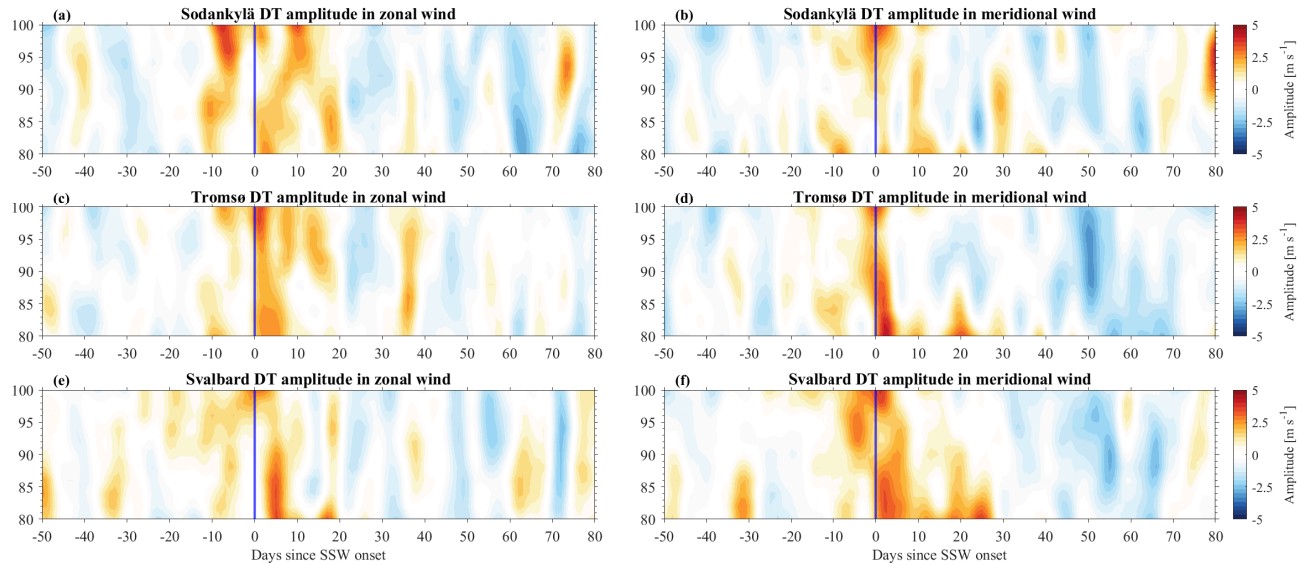

**Figure A3.** The same as Figure. 8 but for WACCM-X(SD).

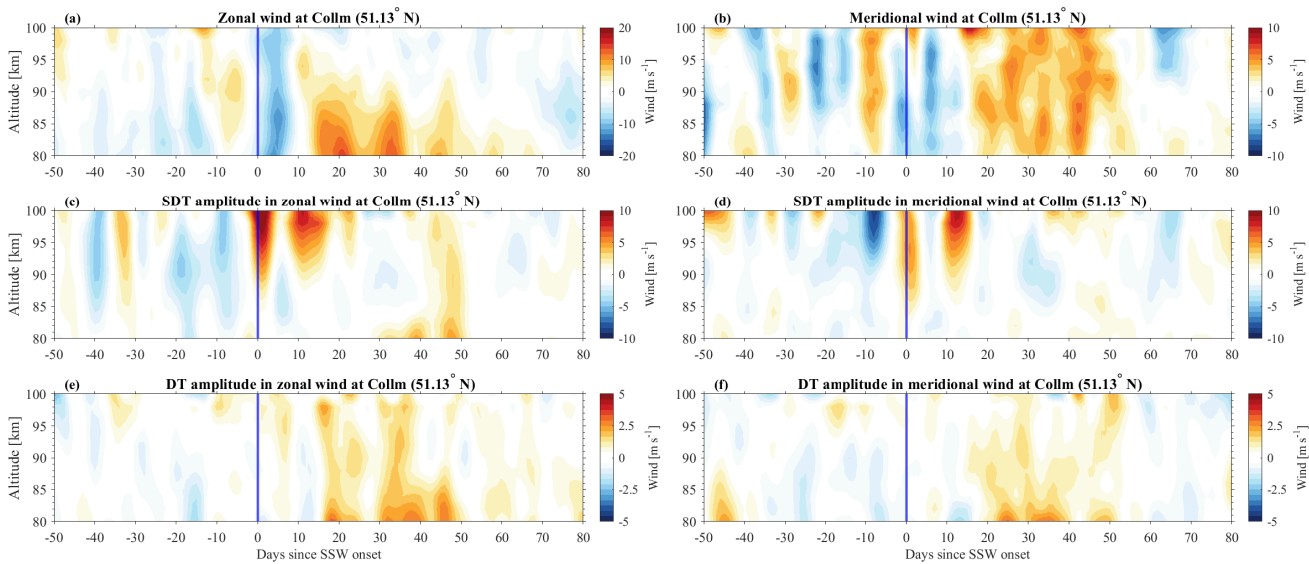

**Figure B1.** The cross-section of SSW composite wind anomaly, SDT and DT amplitude anomalies in the zonal (left panels) and meridional (right panels) wind components observed with MR at Collm (51.13°N, 13.01°E).

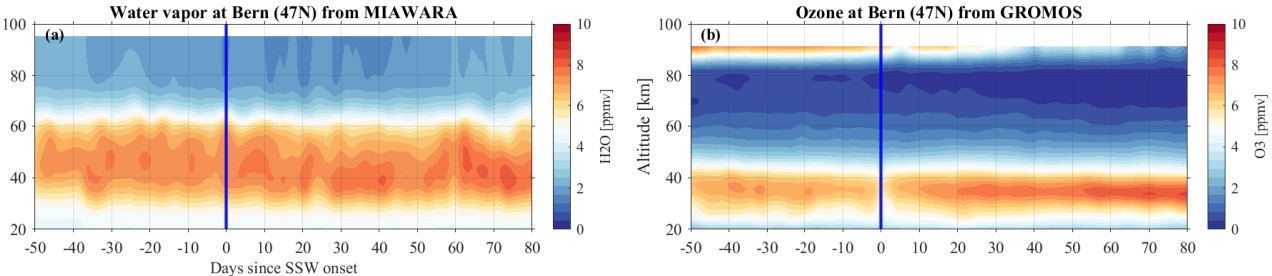

**Figure B2.** The cross-sections of SSW composite water vapor and ozone from ground-based microwave radiometers GROMOS and MI-AWARA at Bern (47°N, 7°E).

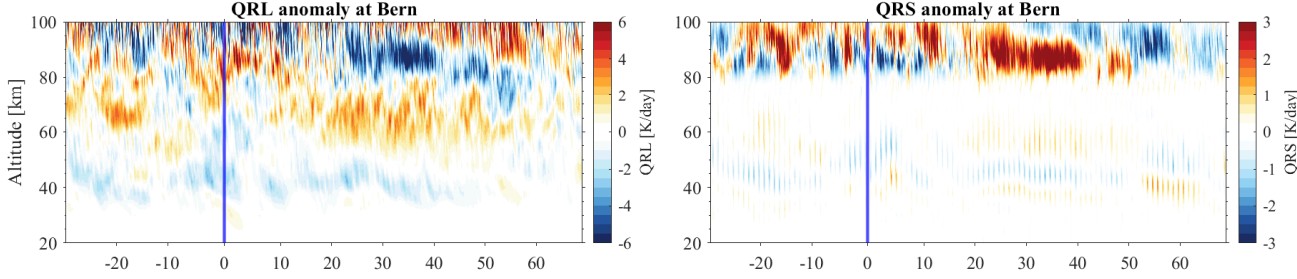

**Figure B3.** The cross-sections of SSW composite QRL and QRS anomalies from WACCM-X(SD) at Bern (47°N, 7°E).

## Appendix C:  Dynamic effects

To better understand these processes, we calculate dynamical heating and cooling rates associated with the air motion in the Arctic region (65-90°N), along with the long-wave cooling and short-wave heating rates from a diagnostic viewpoint, as shown in Figure C1. The dynamic rates associated with meridional and vertical motions according to the transformed Eulerian mean framework described by equations 3.5.1 in Andrews et al. (1987) are calculated as follows:

$$Q_{dyn} = -[\overline{v}^* \frac{\partial \overline{T}}{\partial y} + \overline{\omega}^* (\frac{H N^2}{R} + \frac{\partial \overline{T}}{\partial z})] \tag{C1}$$

where $\overline{T}$ is the zonally averaged deviation from the global mean temperature, $\frac{\partial \overline{T}}{\partial z}$ represents the temperature gradient, and $\frac{H N^2}{R}$ represents the global mean static stability. $Q_{dyn}$ is the dynamic heating/cooling rate. The $\overline{v}^*$ and $\overline{\omega}^*$ are the residual meridional and vertical winds.

The results indicate that anomalous vertical descent, as shown in Figure C2, causes adiabatic heating in the stratosphere around 10 hPa before the SSW onset, which is only minimally offset by QRL radiative cooling, making the process nearly adiabatic. The upward branch of the enhanced circulation, on the other hand, leads to dynamical cooling in the mesosphere, minimally offset by a reduction in QRL. Although the dynamical heating/cooling term $Q_{dyn}$ dominates in the stratosphere and mesosphere (Figure C1a), the radiative effects from QRL and QRS (Figure C2b, c) play a crucial role in shaping the evolution of the thermal structure of the middle atmosphere. The residual vertical wind is calculated at each time step using Equation 3.5.1b in Andrews et al. (1987). In the middle stratosphere before day 0, a negative $\overline{\omega}^*$ indicates downwelling, resulting in dynamic heating due to enhanced descent. In the mesosphere at onset, a positive $\overline{\omega}^*$ signifies upwelling, leading to dynamic cooling associated with increased ascent.

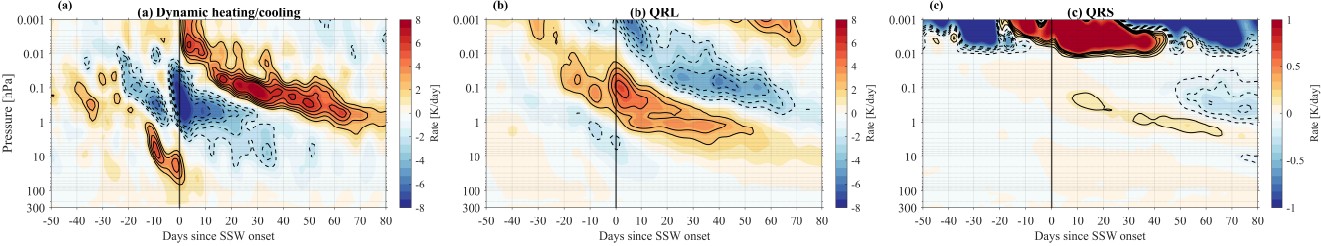

**Figure C1.** The cross-section of the anomaly of dynamic heating/cooling ($Q_{dyn}$), QRL, and QRS heating rates averaged over 65-90°N from WACCM-X data.

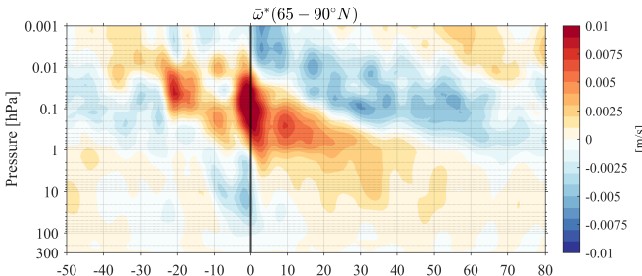

**Figure C2.** The cross-section of $\overline{\varpi}^*$ anomaly during SSW compiste.

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
