# Peer review of "New insights into the polar ozone and water vapor, radiative effects, and their connection to the tides in the mesosphere-lower thermosphere during major Sudden Stratospheric Warming events"

_EGUsphere, 2024_

## Author Comment (AC1)

**Response to referee#1's comments for manuscript**

We'd like to thank the Reviewer for his/her positive feedback and valuable comments. Here we clarify the comments of the Reviewer, with his/her comments in black and our responses indented in blue. To clarify the results we introduced the following changes:

- a case study comparing WACCM-X(SD) and all observations plus MERRA2

- all figures are now provided in altitude

- we added sections comparing ozone and water vapor with WACCM-X(SD)

- identical tidal analysis for WACCM-X(SD) and meteor radars.

This study presents the analysis of observed high-latitude tidal and trace gas (ozone, water vapor) variability in response to SSWs, in combination with modeled trace gas and heating rate variations. The authors seek to leverage the simulated heating rates to quantify the impact of ozone and water vapor radiative perturbations on tidal variations in the MLT. By highlighting the role of trace gases, the authors aim to present the first comprehensive attempt to explain mesospheric tidal variability in the polar region. While the interpretation of the results is built upon a well-researched literature framework, the connection between known mechanisms of tidal variability and the result of the current work is done qualitatively.

Since the main aim of the paper is to extend previously published results on ozone and water vapor coupling to mesospheric tidal variability, by specifically quantifying the contribution arising from polar trace gas variations, my main concern is that, in my view, this relationship is not in fact quantified. The paper remains at the stage where observations are argued to be explained by a possible mixture of dynamical and radiative effects, falling short of an actual quantification of the isolated radiative effects that are central to this work. I am further concerned that the relationship between SSW-driven ozone perturbations (and the corresponding radiative effects) and tidal variations might be hardly quantifiable, due to the lack of sunlight at these latitudes (upwards of 65 degrees North) during wintertime when SSWs occur. While the authors demonstrate that stratospheric and mesospheric altitudes receive some sunlight at these latitudes (even though the exact dates of the individual SSWs are not specified in the text; with some presumably occurring during the polar night), the modeled 0.5K/day perturbation is undoubtedly of very low intensity. While some context is provided for the modeled 0.5 K/day heating rates, the actual tidal forcing components of these rates are not discussed.

To support the author's conclusion that this work provides a deeper understanding of the mechanisms driving tidal variability during SSWs, I therefore think that as a first step, the tidal forcing terms should be much more rigorously quantified. But even then, I would be surprised if the tidal forcing will be significant enough to result in an observable perturbation, given also the large number of other mechanisms and forcing terms involved in driving tidal variability during SSWs.

Given the tools used by the authors, possible suggestions for quantifying the tidal forcing perturbations could be, for example, to repeat the SD-WACCM-X experiments with specified ozone concentrations (preferably only past 65 degrees North), and/or to calculate Hough mode tidal forcing terms (or even simply 24, 12, and 8 hour tidal harmonics at the radar sites) from the 3-hourly SD-WACCM-X heating rates.

1) **Regarding the explanation of a mixture of dynamical and radiative effects:**

In the revised manuscript, we have focused on the radiative effects of trace gases by analyzing 3-hourly variations. We have clarified that the delayed enhancement of the diurnal tide is primarily related to the radiative effect, as evidenced by the strong correspondence between the diurnal ozone cycle and the shortwave heating rate variations at high-latitude stations. Additionally, we have clarified that the enhancement of the semidiurnal tide during SSWs is linked to the secondary ozone layer above 90 km. In

addition, the ozone double-layer structure in the WACCM-X(SD) ozone anomalies that form at the onset of the SSW and last for about two weeks, results in two layers of substantial UV heating. The superposition of these two diurnal tidal waves at the mesosphere may effectively amplify the SDT at high latitudes due to a 12-hour phase offset caused by the different vertical distances both waves have to travel, considering the typical vertical wavelengths of 30-50 km for semidiurnal tides at this latitude.

**2) Regarding the concern about the limited sunlight at high latitudes during wintertime:**

We have presented shortwave heating rates from 3-hourly WACCM-X(SD) simulations during the 2018/2019 SSW event as a case study (with a central SSW date of January 2, 2019). Our analysis reveals a clear diurnal forcing signature in the shortwave heating rate (QRS) approximately 20–30 days after the SSW onset. This response is primarily driven by the modified ozone volume mixing ratio (VMR) in the stratosphere. These results demonstrate that, despite reduced sunlight during winter at high latitudes, stratospheric altitudes still receive sufficient solar radiation to modulate shortwave heating rates, contributing to diurnal tidal variability. We also added one table to show the calculated altitude during winter at three stations.

**3) Regarding the exact dates of the individual SSWs:**

Our study focuses on major SSW events occurring in mid-winter. The WACCM-X(SD) simulations span 2015–2023 and include four SSWs (February 12, 2018; January 2, 2019; January 3, 2021; and February 16, 2023). The meteor radar analysis covers a broader period from 2004 to 2023, encompassing 10 major SSWs, all occurring in January or mid-February.

**More general comments are given below:**

The abstract mentions that the water vapor perturbations have a negligibly small impact on tidal changes. As a suggestion, I would therefore consider changing the title to "New insights into polar ozone and water vapor, and the connection between ozone radiative effects and tides in the mesosphere-lower thermosphere during major sudden stratospheric warming events". I would also consider changing line 19 to read "...transport of radiatively active ozone is important for explaining the observed tidal variability", rather than "transport of radiatively active gases is important…".

Reply: Thank you for the suggestion. We have decided to keep the role of water vapor in the abstract and title, as it plays a part in the overall analysis. The revised Figures clarify the impact of the trace gases as proxies for the QRL and QRS.

L6: 'polar latitude' to 'polar latitudes'.

Changed

L10: 'TDT tide' to 'TDT amplitudes'.

Changed

L50-51: I would suggest adding information that the intensified tides and quasi-two-day wave amplitudes observed by Lima et al. (2012) were observed in the low-latitude summer hemisphere.

Added

L76-79: The authors note that this study, for the first time, quantifies the impact of ozone and water vapor responses on tidal variations in the MLT. Do the authors mean to say the ozone and water vapor responses specifically at polar latitudes? The impact of ozone perturbations on MLT tidal variations has been investigated in numerous other studies, as also referred to in the introduction. It is also not clear to me how the paper

presents a quantification of the total radiative forcing during SSW events; maybe total radiative forcing above 65 degrees North is meant, based on the results showing averaged data between 65-90 degrees North?

Reply: Yes, we specifically investigated the ozone and water vapor responses at polar latitudes. While previous studies have examined the effects of ozone on tides in the MLT, they primarily focused on tropical regions. To our knowledge, no studies have investigated the combined effects of ozone and water vapor on tidal variations in the MLT at polar latitudes during SSW events. Regarding the quantification of radiative forcing, we refer to the total radiative forcing above high-latitude and polar stations (67°N, 70°N, 79°N). Our analysis is based on the shortwave heating rate and ozone anomalies from WACCM-X(SD) simulations at these three stations during SSW events. The total radiative forcing (QRL) is primarily driven by ozone VMR (accounting for approximately 99% in the stratosphere, as shown in Figure 4.24 of Brasseur and Solomon, 2005). Therefore, we quantified the radiative effects on tidal amplitude during SSW.

L144: Perhaps it is enough to mention that SD-WACCM-X can be run in a fully coupled mode. Currently it is not clear whether the model is run with an active or prescribed ocean, which in a way distracts, since this is not strictly necessary information for understanding the results of this work.

Removed

L152: I would suggest rephrasing "photochemical rate constants" to "photochemical absorption and quantum-yield data", or alternatively "photochemical molecular data" since currently it could be interpreted as if the photochemical reaction rate coefficients (J-values) are kept constant, but clearly these vary as a function of solar zenith angle and trace gas concentrations.

Changed

L154: Could it be specified which years exactly the climatological SD-WACCM-X simulations span? Given that short-wave radiation absorption is central to the paper, could the (E)UV absorption scheme and relevant chemical species from WACCM-X also briefly be discussed?

Added: WACCM-C(SD) simulation is from 2015 to 2023.

L169-L170: A table showing the central SSW dates between 2004-2022 would be highly beneficial, given that insolation can be very different between an event occurring on the 24th of March (in 2010) or 5th of January (in 2004). Could it also be specified which model from the cited NOAA page is used as reference (even though I would think this is MERRA-2 based on its relation to SD-WACCM-X)? It would also be helpful if the exact SSW onset criterion is specified in the text, given the large variety of definitions used in literature.

Reply: The SSW events are already presented in a table in our paper, which we have cited accordingly. In this study, we focus specifically on major SSWs that occur in mid-winter (January and February) and do not consider events with onsets such as March 24th, 2010. Generally, SSWs occurring in early spring are referred to as final stratospheric warming (FSW) events rather than major SSWs. Additionally, we confirm that the reference model from the cited NOAA page is MERRA-2.

L174-177: To my knowledge, not all major SSW events are associated with an Elevated Stratopause (ES). Looking at the NOAA database, there are 8 major SSWs between 2004-2013. However, Limpasuvan et al. (2019) identified only 5 ES SSWs during this time period (their Figure 1). Please discuss the notion of not all SSWs having an ES, and how this affects the interpretation of your results, as ES SSW characteristics are assumed to be present as a general feature in the modelled and measured composite SSW response later on in the text.

Reply: here, we have clarified the distinction between major SSWs and Elevated Stratopause (ES) SSWs. A major SSW is defined by two criteria: A reversal of the zonal-mean zonal wind at 10 hPa (∼30 km) and 60°N

from westerly to easterly. A significant increase in stratospheric temperatures, particularly in the mid-to-upper stratosphere. Major SSWs occur predominantly in winter and are classified into displacement and split events based on the vortex morphology.

Elevated Stratopause (ES) SSW is a specific type of major SSW that leads to an exceptionally high and rapidly reformed stratopause after the event. While many major SSWs are associated with elevated stratopause, not all exhibit the pronounced mesospheric and lower thermosphere effects seen in ES-SSWs. ES-SSWs often have a stronger and more prolonged impact in the mesosphere and thermosphere due to enhanced gravity wave forcing and residual circulation changes.

L180: The link between planetary-wave activity and the observed oscillations in the meridional winds is unclear to me. While I understand that planetary wave-mean flow interactions can induce low-frequency oscillations in the mean meridional winds, the analysis described in Section 2.1 states that the time-frequency analysis also includes longer period waves such as stationary planetary waves, in addition to a mean wind. How is it possible to differentiate between low frequency (quasi-stationary) planetary waves and mean wind oscillations at a single station without additional longitudinal information?

We are analyzing local observations in the European sector. The zonal and meridional mean winds, which we obtain from the adaptive spectral filter represent a low pass filtering corresponding to a daily mean. These low-pass filtered winds, therefore, show signatures of planetary wave activity. Planetary waves show stronger and more clear signals in the meridional components as the mean (temporal) meridional wind is close to zero. Our analysis is not separating quasi-stationary waves from the mean flow. However, previous studies have shown a close correspondence of the planetary wave activity in meteor radar winds with global satellite data (https://doi.org/10.1016/j.jastp.2011.10.007,www.atmos-chem-phys.net/18/4803/2018/, https://doi.org/10.5194/angeo-35-711-2017).

L188: At this point naming the station coordinates becomes a bit repetitive. My suggestion would be that here "the three radars" is simply enough.

Changed

L192: The interpretation of the %-changes in tidal amplitudes and mean winds would benefit from a comparison to climatological values. Could the climatological values between for example January and March be included as figures in the appendix? And similarly for the long- and short-wave heating rates? Given the spread in SSW onset dates (4th of January to 24th of March), I would expect there to be a considerable spread in event-to-event QRS rates and the associated deviations from climatology, given the rapid fall off of insolation during winter. It would probably also be helpful to discuss the event-to-event variability in QRS in the paper, based in part on the climatological figures.

We clarify that the onset dates considered in this study are in January and February, as we focus on major SSWs and do not include final stratospheric warming (FSW) events that occur in early spring. The climatological values at these stations have been previously published (Stober et al., 2020). While insolation during winter decreases significantly, it can still reach the stratosphere, troposphere, and even the surface. For reference, please see the added Table 1 in the revision.

L195: Possible reasons for the observed TDT variations are suggested here. Would this be more appropriate for the discussion section? Alternatively, I would suggest moving certain discussion points, for example that the TDT may be contaminated by gravity waves in the analysis technique, to this section. With this in mind, could the authors comment on why the TDT enhancement 10 to 20 days after the SSW onset is so sporadic? For example, at the Tromsø site, the meridional anomaly is around + 2 m/s on day 30, then falls to zero, and then returns to + 2 m/s on day 40. How does this variability fit in with the kind of fluctuations that could be expected from GW contamination?

We put less focus on the TDT variations and moved all details to the discussion. Due to the short window used in the adaptive spectral filter, TDT amplitudes and phases can be biased by gravity waves that fall into the filtering window, which is the case for some inertia gravity waves. Some of the variability can be explained by gravity wave source variability and is not necessarily related to TDT tidal short-term changes.

L201: Please be more specific about how exactly the TDT variability observed in the current work aligns with previous studies that used GCMs and satellite measurements to discuss the solar heating, nonlinear interactions, and gravity wave-tide interaction excitation mechanisms for the TDT. This also ties in with the above comment.

A detailed discussion of the gravity wave and TDT tidal variability is beyond the scope of the paper and requires additional work on the methodology. Separating gravity waves and terdiurnal tides is challenging from point source data. Currently, we are working on tomographic meteor radar network analysis to improve our tidal diagnostic. However, as the Hunga Tonga eruption did show, our available domain size is still within the range of large scale gravity wave perturbations and, thus, there will be some ambiguity remaining. However, this would go beyond the scope of the paper.

L200: Do the authors here mean to say "TDT amplitude anomaly"?

Changed

L213: Personally I do not see a clear sign of the STD showing an enhancement 20-50 days after the SSW onset (albeit weaker than during SSW onset time), at least not in the data presented in the current work. In the current work the STD shows considerable variability also during times without SSWs, while there is no commonly identifiable pattern between the three meteor radar stations between days 20-50. If anything, I would argue that only Sodankylä shows a local maximum between days 40-50.

Removed

L234: It is not clear to me what Figure 5 shows. From the context I would guess that these are daily averaged 3-hourly heating rates? Or are they amplitudes of the 24 hr variations in the (3-hourly) modeled heating rates? This is crucial information for understanding discussion points later on.

Yes, Figure 5 presents the daily-averaged 3-hourly heating rates. In the revision, we have updated this figure to show 3-hourly heating rates instead of daily averages to better capture the variability. Furthermore, the new Figures clearly show the diurnal cycle in the QRS and ozone VMR.

L254: This is largely a repeat from an earlier comment, but I fail to see a clear SDT enhancement at the three stations 20-50 days after SSW onset. It is therefore difficult to connect the modeled QRS rates to SDT amplitude anomalies 20-50 days after onset. As a suggestion, could a time-series of, for example, 90 km altitude SDT amplitudes and 50 km altitude QRS rates be shown in a single figure? This may help to demonstrate a more clear relationship between the two.

Reply: In the revision, we have clarified that the SDT enhancement is primarily associated with the secondary ozone layer in the thermosphere, above 90 km, which aligns with the observed SDT enhancement at these altitudes. Stratospheric ozone layer (20-50km) plays a key role in the delayed DT enhancement. QRS is mainly contributed by the ozone layer UV absorption. This double-layer structure is mainly responsible for the semidiurnal tidal activity during the winter months at these latitudes. The superposition of the in-situ forced tide at the secondary ozone layer is out of phase with the stratospheric tide excited around 50-60 km altitude that needs to propagate upward 30-50 km. This coincides with the vertical wavelengths that are found in climatological analysis (Stober et al., 2021b) for these stations. During the SSW, this balance is disturbed, and the secondary ozone layer together with the elevated stratopause provides favorable conditions to amplify the SDT. The QRS shows a pronounced enhancement that coincides with the time of the semidiurnal tidal amplification.

L255: I do not quite follow the line of reasoning where 1) the findings from Siddiqui et al. (2019) and Limpasuvan et al. (2016) that SSWs are followed by rapid increase of ozone between 20S to 40N arising from equatorial upwelling and cooling, and a decrease poleward of 40N, and 2) the subsequent ozone enhancement at mid to high latitude as shown in Figure 6 happening immediately after SSW onset. Based on the first point, I would expect a rapid decrease in ozone centered on the SSW onset date in Figure 6?

Reply: We have added the ozone variation at the mid-latitude station in Bern (47°N) in Figure B2. As expected, there is a decrease in ozone at this station around the SSW onset. At polar latitudes, SSW events, characterized by abrupt warming and the weakening or reversal of the polar wintertime westerly circulation, lead to extreme ozone variability. This allows ozone-rich air from lower latitudes to enter the polar region, contributing to the observed ozone enhancement in the polar stratosphere shortly after the SSW onset.

L256: I would suggest adding that these heating rates are for wintertime. I would further expect these heating rates to be quite different between January and March SSWs, given the high latitude of the stations, and considering that heating rates approach 14 K/day during summer (Brasseur and Solomon, 2005, Figure 4.25). As mentioned above, I think it would therefore be highly beneficial if climatological heating rates between, say, January and March could be added to the appendix.

Please see below: reply with L257.

L257: It is argued that the QRS change is mostly due to ozone increases following the SSW onsets. The QRS anomaly falls roughly between 0.1 to 1 hPa (Figure 5), while the ozone anomalies extend between roughly 50 to 0.1 hPa in two largely separate patches (Figure 6). Why wouldn't the ozone anomaly below 1 hPa, i.e. the bottom patch, contribute to the QRS anomaly? Is this because the stratosphere does not receive any sunlight at these latitudes during winter? A comment at this point in the manuscript would be beneficial. The lack of QRS perturbation between 100-10 hPa seems to conflict with the stratosphere receiving sunlight as argued for in section 3.3.

Reply L256 and L257: We have added the absolute values of heating rates in Figure 15, where the QRS rate reaches up to 15 K/day after the SSW onset, highlighting the significant variation in QRS heating rates. This addition helps illustrate the variability in heating rates during wintertime SSW events. The stratosphere does receive sunlight at these stations during winter, as discussed in Section 3.3. For further clarification, please refer to Table 1 in the revision.

L286: Could the double-layer structure of the ozone be discussed or clarified in more detail? Does this refer to the anomaly in Figure 6 appearing to show a two-layer structure? I would think that a two-layered anomaly would not necessarily imply a double-layered structure of the underlying layer. Could the upper anomaly (centered on 1 hPa, or 50 km) simply be an extension of the stratospheric ozone layer?

Reply: we have clarified that the double-layer structure is the stratosphere ozone layer (20-50km) and secondary ozone layer (90-100km).

L286-300: It is hard to imagine a 0.5 K/day heating rate (if these indeed would amount to 0.5 K/day) to provide significant tidal wave energy. In my view, isolated model experiments and a more detailed discussion of the tidal forcing terms would be a necessary step here. As a first estimate, what is the amplitude of the 12 hr component in the heating rates?

Reply: We added one section: a case study during SSW 2018/2019 to show the heating rate variation, which reveals a clear signature of a diurnal forcing starting 20-30 days after the SSW.

The stratospheric diurnal tide is further a largely vertically trapped mode, so it is unclear to me how the amplitudes at stratospheric altitudes (40-50 km) translates to amplitudes at meteor radar altitudes (80-100 km). The superposition of two 24 h waves will also always result in a 24 h wave, no matter how in or out of phase

they are (so long as their frequencies are both 24 h). So it is also unclear to me why the superposition of two diurnal waves may effectively amplify the SDT at meteor radar altitudes? Regardless, I think the impact of travel time and vertical distance should be explained in more detail.

Reply: The ozone double-layer structure in the WACCM-X(SD) ozone anomalies that form at the onset of the SSW and last for about two weeks, results in two layers of substantial UV heating. The superposition of these two diurnal tidal waves at the mesosphere may effectively amplify the SDT at high latitudes due to a 12-hour phase offset caused by the different vertical distances both waves have to travel, considering the typical vertical wavelengths of 30-50 km for semidiurnal tides at this latitude Stober et al., 2021b, 2020).

L298: I can't find reference to the (in-situ) diurnal heating rates causing pronounced diurnal tides at the latitude and altitudes relevant to the current work based on the cited work by Schranz et al. (2018), which appears to discuss only the diurnal cycle in ozone and not the winds. Could this be clarified?

Reply: In the stratosphere, there are strong diurnal heating rates, as shown in Figure 2, primarily contributed by ozone. Schranz et al. (2018) highlighted the diurnal cycle of ozone. However, the diurnal heating rates due to ozone give rise to the formation of diurnal tides.

L310: Here the observed SDT response is primarily attributed to changes in zonal mean wind and ozone heating at mid-to-low latitudes, even though the contributions of these effects are not quantified, and these are therefore difficult to place into context with the ozone mechanism described in the paper. The double-layer ozone structure is further argued to contribute to the immediate STD response on line 315, while on line 312 it is argued to likely contribute to the observed STD variability during the recovery phase, and not during the onset phase. This seems contradictory.

Reply: We have clarified that the secondary ozone layer above 90 km plays a key role in the SDT amplitude.

L309-L324: I struggle to see the connection between the discussion points in this paragraph and the placement of the results within the literature. I think this largely stems from the contributions of the different mechanisms (propagation conditions, mid-to-low latitude ozone forcing) not being quantified in the context of the observational data, even though observed characteristics are attributed to these mechanisms. When the aim of the paper is to quantify the contribution of polar trace gas perturbations, I think a more careful quantification or discussion of the other effects is also warranted, given that the net observed tidal response is shaped by the complex interplay of all the different mechanisms. Further, as mentioned above, while the modeled 0.5 K/day heating rates fall within the range of stratospheric diurnal temperature variations, the actual diurnal components of the heating rates are not discussed.

Reply: In the revision, we have clarified the following points: The delayed DT enhancement is associated with diurnal heating forcing, primarily resulting from ozone UV absorption at high and polar latitude stations. The SDT enhancement near the SSW onset is linked to the secondary ozone layer, and the double-ozone layer also plays a significant role in the SDT amplitude. The interaction of these two diurnal tidal waves in the mesosphere may effectively amplify the SDT at high latitudes. This is due to a 12-hour phase offset caused by the different vertical distances both waves travel, considering the typical vertical wavelengths of 30-50 km for semidiurnal tides at this latitude.

Please check the spelling of Tromsø in the figure sub-titles (sometimes spelled as Tromose).

Changed

---

## Author Comment (AC2)

**Response to referee#2's comments for manuscript**

We'd like to thank the Reviewer for his/her positive feedback and valuable comments. Here we clarify the comments of the Reviewer, with his/her comments in black and our responses indented in blue.

The authors would like to thank the substantial comments and suggestions from the referees, which significantly helped improve the quality of this manuscript. We have revised the manuscript carefully based on the comments and suggestions of the reviewer. More details of the revision can be found in the revised manuscript as well as the point-to-point response as follows (all authors' responses here are in blue). However, to clarify the results we introduced the following changes:

- a case study comparing WACCM-X(SD) and all observations plus MERRA2
- all Figures are now provided in altitude
- we added sections comparing ozone and water vapor with WACCM-X(SD)
- identical tidal analysis for WACCM-X(SD) and meteor radars

The study by Shi et al utilized observational data of winds, ozone and water vapor at high northern latitudes to analyze the response of tidal wind amplitudes to sudden stratospheric warmings (SSWs), and ask the question whether anomalies in radiative active trace gas abundances may contribute to the anomalies in tides. Global model data are used to deduce anomalies in radiative heating rates before, during and after SSWs. The study presents interesting signals in tidal amplitudes of the diurnal, semi-diurnal and terdiurnal tides at three different latitudes. Furthermore, trace gas anomalies (water vapour and ozone) associated with SSWs are shown to be very consistent between satellite data (MLS), global model simulations (WACCM-X) and local measurements in Svalbard, which is a very encouraging result. The consistent and comprehensive quantification of tidal and trace gas anomalies around SSWs from observational data is a valuable set of analysis worth publication. The study further puts forward the suggestion that the trace gas anomalies are important in "explaining the observed tidal variability during SSW events" (abstract, line 19-20). While I agree that the modification of tides by ozone anomalies is a plausible mechanism, I have to disagree that the pieces of analysis shown in the study provide any quantification of this radiative effect, as detailed below. It is a valid and interesting point to discuss the possible impact of the trace gas anomalies (and associated heating anomalies diagnosed from the model) on the tidal anomalies, but the authors should present those points as discussion of possible (!) effects, rather than stating that a comprehensive explanation of tidal variability is obtained in the study. Furthermore, there are a number of corrections necessary in to improve the presentation of the results, and at a few places I found the description of the results inconsistent. Overall, I recommended that the authors revise their manuscript majorly, focusing the study on the quantification of SSW signatures in tides and tracers.

**Major comments:**

1) Quantification of role of trace gas anomalies to force tidal anomalies

The authors state that they "presents the first comprehensive attempt to explain mesospheric tidal variability ..." (line 306, similar in abstract line 19-22). As stated above, I generally agree that the study presents an interesting and comprehensive quantification of tidal variability and co-located tracer variability from observational data, and it is valid to discuss whether there could be possible effect of tracer anomalies on tidal variability. However, I have to disagree that the study is a comprehensive attempt to explain the tidal variability, for which an actual quantification of the role of trace gas anomalies for the tidal generation would be necessary.

While the presentation of the radiative heating rates from model data help to infer whether there might be an effect at all from the tracer anomalies, this is not sufficient to conclude on whether the tracer anomalies actually

play a role to cause the tidal anomalies. For this, as least two more steps would be necessary: a) quantify how much of the radiative anomaly is actually due to the tracer anomalies; b) quantify how much the anomalous radiative heating/cooling contributes to tidal generation. The former could be done by offline radiative calculations. A full quantification would, as far as I can see, only be possible by conducting model simulations which either include or discard the tracer anomalies. As of now, the statements made in the paper on the role of tracer anomalies for tidal variability are at most based on scaling arguments (e.g., role of tracer anomalies for heating, see comment on lines 257-259; argument about similar strength of relative anomalies in line 365-366), and many very speculative statements are made in particular on the relative role of anomalous heating/cooling via tracer anomalies versus anomalous propagation due to changed background winds (e.g., on page 15, see individual comments below on line 257-259, line 284, 286).

I understand that conducting simulations for the actual quantification would go beyond the scope of the paper. Therefore, I suggest that the authors focus on the quantification of the tidal and tracer anomalies from observations, which by itself composes an interesting set of analysis. This can be complemented by a discussion of possible effects of the tracer anomalies on tides, but making sure to emphasize the uncertainties and the speculative nature of some of the arguments/hypothesis put forward.

We appreciate this detailed feedback and valuable comments regarding the quantification of the role of trace gas anomalies in tidal variability.

a) We have reconstructed the structure of this manuscript, presenting an interesting and comprehensive quantification of tidal variability and co-located tracer variability from observational data, and discussing the possible effect of tracer anomalies on tidal variability.

b) We added a case study to show the heating rate variation, which reveals a clear signature of a diurnal forcing starting 20-30 days after the SSW. The heating rate is mainly contributed by the absorption of UV radiation by ozone in the stratosphere and in the lower mesosphere. Therefore, we clarified that the radiative heating rate from ozone plays an important role in the DT amplitude enhancement.

c) WACCM-X(SD) tides are generated by the interactive chemistry and, thus, by the radiative forcing. Water vapor and ozone are presented as proxies for the total balance. The nudging at the lower boundary might also prescribe part of the tides at the stratosphere, however, as shown in the case study a few kilometers above the nudging altitude the WACCM-X(SD) GW parameterizations dominate the dynamics and the radiative forcing from the chemistry provides a key sources for the tidal forcing.

2) Inconsistencies in the discussion on SDT

Much of the discussion on the effects of ozone anomalies on tides focuses on STD, which was shown to be enhanced just around the central date of SSWs. I have to admit that I got confused about the timing of when this link is suggested to act. At places, it is stated that the STD enhancement is linked to the persistent ozone (and associated QRS) anomaly over several weeks after the SSW (e.g., in line 253), but this is inconsistent with the timing of the STD anomaly just around the SSW event. At other places, the STD enhancement is stated to occur at the SSW onset, and that it is "... attributed to zonal wind changes and ozone heating at mid- to low latitudes" (line 311). Firstly, there is no actual attribution presented in this study (see major comment 1 above), and secondly ozone heating at low- to mid-latitudes is not shown here, so this confuses me even further. Please ensure to clarify better the proposed link between ozone anomalies and STD anomalies (see also individual comments below on line 253, 311, 315, 329)

We have added one chapter to illustrate the water vapor, ozone, and radiative effects, linked to the SDT amplitude. WACCM-X(SD) simulation shows the primary stratospheric ozone layer at 20-50 km, the tertiary ozone layer around 60-70 km, and the secondary ozone layer above 90 km. We have clarified that this double-layer structure (stratospheric ozone layer and secondary ozone layer) is mainly responsible for the semidiurnal

tidal activity during the winter months at these latitudes. The superposition of the in-situ forced tide at the secondary ozone layer is out of phase with the stratospheric tide excited around 50-60 km altitude that needs to propagate upward 30-50 km. This coincides with the vertical wavelengths that are found in climatological analysis (Stober et al., 2021b) for these stations. During the SSW, this balance is disturbed, and the secondary ozone layer together with the elevated stratopause provides favorable conditions to amplify the SDT. The QRS shows a pronounced enhancement that coincides with the time of the semidiurnal tidal amplification.

Additionally, we analyzed observations from the southernmost Meteor Radar (51.3N) available in the European sector, located at the edge of the polar vortex and, which sometimes shows signs of SSW events (see Appendix B). As a result, the derived mean winds and tidal anomalies show a morphology similar to that observed at polar and high latitudes. In particular, the enhancement of SDT remains evident. A few degrees further south, observations from MIAWARA and GROMOS provide additional insights. The water vapor and ozone VMR for the location at Zimmerwald/Bern (47N), Switzerland. Performing the same epoch analysis used for GROMOS-C and MIAWARA-C exhibits no longer a signature of the SSW events. This is also reflected in the QRL and QRS anomalies shown in Appendix B. The QRL and QRS cooling and heating rates indicate a more stratified vertical structure resembling the layering found in the trace gas measurements at the stratosphere and lower mesosphere.

3) Methodological issues and presentation of results

- Some methodological clarifications are needed, e.g. stating more clearly which time periods are used for the different data sets and clarifying some details on how the tides were fitted and how the (relative?) anomalies were calculated. See individual comments below.

We have more clearly stated the time periods used for the different data sets and elaborated on the fitting process for the tides. We also have clarified the details of how the (relative) anomalies were calculated.

**Individual comments:**

Title: I suggest to revise the title; in its present form there might be a word missing ("the polar ozone and water vapour ANOMALIES (?)"), and the "radiative effects" in the middle of the title seems out of place. Also, is "polar" referring to everything, or just ozone (as it appears right now)? How about simplifying the title to something like "Response of tides in the polar MLT to SSW events, and their link to ozone and water vapour anomalies."

All stations used in the current study are located north of the polar circle and, thus, we plan to keep the term polar in the title. Furthermore, we present more details leveraging WACCM-X(SD) fields at a sub-daily resolution to visualize the diurnal effects of short-wave absorption and ozone chemistry. We also provide similar viewgraphs for water vapor, although less relevant for the direct tidal excitation. The QRL is mainly contributing to the background wind and temperature fields in which tides propagate. In so far, we decided to keep the title.

**Abstract**

line 7: "polar ozone and water vapor in linking mesospheric tidal variability...": what is "linking" referring to here? maybe you mean "driving", or "contributing to" (but those words would be too strong, see major comment 1). Please rephrase.

Changed: driving

line 8-14: The description of anomalies in tidal amplitudes appears rather clearly written up until line 12, but misses to clarify which wind component the text refers to. Starting in line 11, a distinction is made between

zonal and meridional wind components, making me wonder which component was refers to in the earlier sentences? Please clarify.

Changed: a significant negative anomaly in TDT amplitudes in zonal and meridional components is observed.

line 19-22: "The interaction between dynamic processes and the transport of radiatively active gases is important for explaining the observed tidal variability during SSW events": this is not backed up by the results, see major comment.

Removed

**Section 1**

- line 48: are the "modified zonal mean zonal winds" modifying tidal amplitudes via propagation of tidal waves? Please clarify and be more specific on the suggested mechanisms from literature.

Changed

- line 51: "heightened planetary wave activity" - change to "strengthened planetary wave activity"

Changed

- line 54: did the authors meant to distinguish between mixing ratio and ozone density within this sentence?

- line 61: "ozone dynamics": consider changing to "ozone transport", not sure what "ozone dynamics" are.

Changed

- line 68: I'm not familiar with all the referenced studies, but at least the last three references (Oehrlein et al, de la Camera et al and Hong and Reichler) do NOT analyse tides in the MLT region; they analyze the impact of interactive ozone on stratospheric dynamics including planetary waves. Thus, they cannot serve as appropriate reference for the statements made here.

Added the appropriate reference for the statements made here.

- line 76: QRS and QRL are defined in the abstract, but it would be good to repeat the definition here; also it might be more appropriate to just say radiative heating and cooling here rather than using the variable abbreviations.

Changed

- line 76: it would be good to mention at this point that comparison of water vapour and ozone anomalies from WACCM-X to observations are performed in a first step to ensure consistency with the observational data.

Added sentence: In the first step, we compare water vapor and ozone anomalies from WACCM-X with observations to ensure consistency with the observational data before analyzing their role in tidal variability.

- line 83-84: this description of the structure of the paper is completely generic; it might as well be dropped, or better explain what is done in which section within the previous paragraph.

Dropped

**Section 2**

- line 87: "at three different high latitudes"; consider changing to "at three stations in high latitudes"

Changed

- line 88: continuously in operation in which time period? Please specify.

Specified

- line 90: unclear what is meant by "the same wind retrieval algorithm" - the same over time, between the stations, or the same than in the reference? please clarify.

Clarified

- line 93-94: "tides ... are obtained" - does this mean that the given equation is fitted to the data to obtain the tidal amplitudes? Please be more specific how the equation is used to extract tides.

The adaptive spectral filter is much more complicated than the equation is expressed, although equation (1) is presenting the correct kernel function. The tidal fit is carried out with an adapted window length for each tidal frequency (diurnal tide- 48 hours, semidiurnal tide – 24 hours, terdiurnal tide – 16 hours). We have carried out several tests to optimize the oversampling factor. Furthermore, we included a Gaussian weighting to avoid window effects as they are known for Fourier effects. The solution of the larger windows is used as Tikhonov regularization for the next smaller window for the unfitted tidal components. There is also a constrain about the vertical wavelength of the tide. We mitigate a potential contamination due to the gravity waves by applying a smoothness constrain on the tidal phase (not amplitude). Details are provided by the referenced in the manuscripts. The adaptive spectral filter targets key issues of tidal analysis. The algorithm can handle data gaps and unevenly sampled data, contains full error propagation, and most importantly only requires phase stability with each adapted window length.

- line 98: where are the zonal mean variables obtained from for the analysis of the tides from the meteor radar? Or are they a result of the fitting?

The term zonal mean does not refer to the often used 'zonal mean zonal' wind for model analysis. The 'mean' refers to the daily average background. Zonal and meridional mean winds represent daily mean values.

- line 100: planetary waves (waves is missing)

 Added

- line 106 and line 117: is this the same "QPACK" software, since the reference is the same? If yes, please be consistent in the spelling; if not, clarify.

Yes, changed

- line 132: please provide the approximate altitudes for the MLS data, given that meteor radar observations are provided in altitude coordinates. An interpolation to a consistent vertical coordinate would ease comparisons made later between tidal anomalies and trace gas anomalies.

Changed: Corresponding altitudes are included in the text. Comparisons between tidal anomalies and trace gas anomalies have been interpolated into a consistent altitude:

- line 141: I suggest removing "A comprehensive numerical model.." (both unnecessary and incomplete sentence).

Removed

- line 143-144: "capable of being run ..." consider rephrasing, e.g. simply to "WACCM-X can be run with a coupled or prescribed ...". Generally, I suggest to rather focus on describing the configuration of WACCM-X that is used here, rather than listing possibilities and developments in the CAM/WACCM/WACCM-X framework (this list would be rather long)!

Changed

- line 150-153: I wouldn't agree that the developments described here are "recent" (going back to 2010), nor do I believe they are particularly relevant to be mentioned for the study here.

We agree with that. This sentence has been removed.

- line 155: please rather mention the years of the simulation than saying they are done for the measurement period. Also, are those dedicated simulations performed for this study, or the ones available at: https://doi.org/10.26024/5b58-nc53. If not, what is different?

We have added that the simulation period runs from 2015 to 2023. The simulations used in this study are dedicated runs and differ from the dataset available at https://doi.org/10.26024/5b58-nc53, as the older dataset does not include ozone and water vapor and is only available until 2017.

- line 159: here would be a good place to define the QRL and QRS heating rate variables in detail.

Added: The QRS heating rate is primarily governed by ozone absorption of radiation, while the QRL cooling rate is influenced by the presence of carbon dioxide, water vapor, and ozone.

- General: the information on time period of available observations is largely missing in Section 2. Please add.

Added

**Section 3**

- line 166: please add how many events the composites are based on (so the reader doesn't need to check the compendium for the given period).

Added

- line 174: "altitude of wind reversal": the reversal cannot be seen from the anomalies, consider adding the zero wind contour to the Figures.

Added

- line 181: why would positive meridional wind anomalies be associated with reduced wave activity? Please elaborate.

The polar vortex is reestablished after the SSW as indicated here by the period of intensified westerly winds, the planetary waves are rather weak.

- line 181: please specify in this sentence that the references named here (Dowdy et al and Koushik et al) refer to wind and wave response to SSW in the MLT. As the sentence reads now, it sounds like it refers to the dynamics of the wind reversal in the stratosphere (i.e., the SSW generation itself).

We agree with that. This sentence has been removed.

- line 191: I didn't quite understand how the anomalies in tidal amplitudes are calculates - it says here that they are a "relative change", but the units are m/s; also the phrase "taken as a mean time for the entire showcase period" is not clear to me. Furthermore, it would be very helpful to add a significance test to the anomalies in tidal amplitudes. This way one could infer which anomalies are statistically robust.

The anomaly in tidal amplitudes is calculated as the difference between the tidal amplitude observed during an individual SSW event and the climatological mean tidal amplitude from all years within the same period. The anomaly is obtained by the difference between the climatological undisturbed situations vs all SSW events. We have removed the previous phrasing for clarity. Additionally, we have included the anomalies of tidal

amplitude in a case study of SSW events in 2018/2019, which revealed significant anomalies, particularly in DT and SDT. This case study analysis enhances the interpretation of the results and will be added to the revision.

- line 194: percentage difference relative to the mean value (add "relative to")

Added

- line 243: "This indicates...": it is not clear to me what is implied here: that the persistent ozone anomalies are arising from a persistent alteration of the thermal structure (via dynamics), or that the ozone anomalies themselves might give rise, or a least contribute to the persistence of temperature anomalies?

The persistence of positive ozone anomalies reflects prolonged changes in middle atmospheric circulation following the SSW. Additionally, ozone's radiative effects may contribute to maintaining temperature anomalies, suggesting a potential feedback mechanism.

- line 248: I agree ozone changes are dependent on altitude, but Fig. 6 does not provide information in variations with latitude - for the latitudes shown here, the ozone anomalies are remarkably similar.

We added ozone and water vapor data from a mid-latitude site in Switzerland to confirm the latitudinal differences. Although, the difference between Svalbard and the mainland are small and the general morphology agrees, there is some difference in the ozone VMR.

- line 249: It's not clear to me which altitude, nor process is referred to here: I agree that there might be enhanced tropical upwelling during the SSW event due to pronounced wave driving in the stratosphere; However, enhanced upwelling would reduce tropical ozone in the lower and mid-stratosphere. Please specify which altitudes you refer to here.

We only reference a previous study. Our study does not investigate effects outside the polar vortex.

- line 251: "... a decrease poleward of 40°N. The ozone enhancement ... at mid to high latitudes is consistent...": This confuses me - how is the ozone decrease poleward of 40°N shown by previous work consistent with the enhancement of ozone diagnosed from the observations here? Do you refer to different timings (before versus after SSW)? Please clarify and rephrase.

Reply line 248, line 249, line 251: We have added a figure illustrating the variability of ozone at mid-latitude (Bern, 47°N) in Figure A2a, as measured by ground-based microwave radiometers. This addition helps clarify the differences in ozone behavior at different latitudes and better shows the observed enhancements with previous studies.

- line 253, and general: "ozone increase.. and QRS therein": I agree that changes in ozone should primarily affect the shortwave rather than longwave radiation. Therefore, I am surprised why the authors decided to overlay the QRL contours on the ozone anomalies instead of the QRS fields, which would ease making the link between the tracer changes and the resulting radiative changes.

We have changed the overlaid QRL contours on the ozone anomalies.

- line 253: Here, and at several places I was confused which anomalies are referred to when the "enhancement of STD" is discussed: according to Fig. 3, STD is mainly enhanced just around the SSW event (there is some sign of enhancement later, but very noisy signals including reductions, and I doubt this is significant). However, here the "ozone increase and QRS therein" in the "SSW recovery period (day 20-50)" are suggested to be coincident with the STD anomaly - can you clarify?

The enhancement of the SDT (8 m/s) is most pronounced at the onset of the SSW, particularly when compared to the mean winds (25-30 m/s), resulting in a change of more than 40%. In the revision, we have clarified that

the SDT enhancement is primarily associated with the secondary ozone layer in the thermosphere, above 90 km, which aligns with the observed SDT enhancement at these altitudes. This connection addresses the important role of radiative effects of ozone in driving tidal variability during the SSW onset.

- line 257-259: I agree that it is plausible that the ozone anomalies are a main cause of the QRS anomalies, but it would of course be great if this would be quantified, which would be possible e.g. via offline radiation calculations. As is now, the only way to infer the importance of ozone for QRS is the alignment of anomalies, and the rough approximation saying that "mean heating rate by ozone at those latitudes is around 1 K/day". Even more important, the only argument by which the authors rule out the role of water vapour for radiative anomalies is saying that the 25% anomaly "translates to changes in cooling rate of about 0.05 K/day" (line 259). It is not clear to me what this number of 0.05 K/day is based on, and the same holds for the 1 K/day heating by ozone. Given the argument for the importance of the tracers is almost completely based on those numbers, it needs to be made much clearer where they are taken from, for which altitude and latitude region they are valid, and in how far one can linearly scale the radiative impact with the tracer anomalies (as apparently done here).

The ozone data from the stratosphere ozone layer and secondary ozone layer show clear diurnal cycles, which result in a semidiurnal tide at the MLT and a diurnal tide at the stratosphere/lower mesosphere. WACCM-X(SD) fields are in good agreement with the trace gas measurements. We have clarified more details in the added case study of SSW in 2018/2019 and the discussion part. QRS and QRS are not shown in K/day and with a temporal resolution of 3 hours to provide a visual evidence for the difference in their diurnal structure.

- Figs. 6, 7 and 8: great to see the good agreement between the satellite, model and station data! Possibly this result could be emphasized more?

Added: We have added two subsections discussing the water vapor and ozone anomalies, highlighting the comparisons between satellite, model, and station data.

- line 268ff: I agree it is good to calculate the dynamical-driven temperature tendencies in order to compare them to the radative ones. However, for the purpose of the paper and in the following discussion it was not entirely clear to me what this analysis reveals beyond the fact that dynamical heating/cooling is generally balanced by radiative (in particular long-wave) cooling/heating?

We addressed the importance of calculating dynamical-driven temperature tendencies to compare them with radiative tendencies. Our analysis highlights that while dynamical heating/cooling is generally balanced by radiative (particularly long-wave) cooling/heating, the increase in ozone leads to enhanced short-wave heating. However, the direct in situ temperature effect of ozone increase—i.e., stratospheric warming—is intertwined with other factors, such as anomalous vertical descent (negative anomalies in w), which contributes to dynamical heating and influences the total temperature response. It is the feedback mechanism on temperature change. To improve clarity, we have simplified the discussion on dynamical heating/cooling effects and moved the details to the appendixB.

- line 277: ".. the dynamic process drive the persistence of ozone anomalies...": I agree that this is likely the case, but the temperature tendencies do not necessarily help to explain the tracer anomalies - do you suggest it is the anomalous vertical circulation which drives both temperature as well as trace gas anomalies (the later via vertical advection)? Or do you suggest that the temperatures affect ozone via chemistry?

Changed: both dynamic (including eddy effects and vertical advection) and chemical processes drive the persistence of positive ozone anomalies in the upper stratosphere and lower mesosphere. In contrast, water vapor anomalies are primarily governed by dynamical processes, as the mesospheric lifetime of water vapor is on the order of months/years. We have clarified the distinct mechanisms influencing these two trace gases.

- line 281: "still remains in the Earth's shadow..": this would be a good place to provide details on the altitudes which are reached by sunlight as function of latitude. This is addressed by the schematic Fig. 11 later on; if kept, this schematic and the considerations with it should be mentioned here. However, I suggest replacing the schematic Fig. 11 by a figure showing the amount of sunlight in mid-winter as fct of latitude and altitude; this would help to make this point in a more quantitative manner. It could even be shown as a function of daytime to help to make the point on how / where ozone anomalies might affect tidal amplitudes.

We have added a table to show the calculated altitude at these stations.

- line 284: at this point, it appears very speculative to conclude that the redistribution of ozone contributes to tidal variability. I agree this might appear plausible, but the pure quantification of shortwave heating anomalies does not give any quantitative estimate on how strong the effect on tides might be.

We agree with you in pointing out the speculation that the redistribution of ozone contributes to tidal variability. The shortwave heating rate is primarily driven by ozone absorption of ultraviolet radiation. In the revised manuscript, we analyzed the three-hourly anomalies in radiative heating rates (QRL), replacing the hourly anomalies in the manuscript. QRS reveals a clear signature of a diurnal forcing starting 20-30 days after the SSW, which is related to the increased ozone VMR. This result highlights ozone-induced heating anomalies that contribute to tidal variability, providing new insight into the linkage. Furthermore, we want to note that tides in WACCM-X(SD) are mostly driven by the QRS. Below 90 km ozone plays a crucial role for absolute QRS, above other processes become relevant as well. We establish a more clear relationship between both by direct comparisons.

- line 286: results on ozone double layer: I cannot, or barely, see how the double layer of ozone anomalies is reflected in "UV heating" (i.e. QRS) - do I assume correctly that the authors refer to the change of sign in QRS around 10 hPa, visible as change from light blue to light red in Fig. 5 (right)? If yes, it has to be clarified whether the heating in the lower layer is at all different from zero, as this is not apparent from the Figure.

We added new Figures in geometric height showing the altitude of all three ozone layers and their corresponding QRS rates on the same vertical grid. The double layer structure is one of the most clear results in WACCM-X(SD).

- line 286: "these two diurnal tidal waves": do the authors suggest that the double layer of ozone, and thus possibly shortwave heating anomalies, cause two distinct forcing regions of tidal waves? To me, this result appears to be extremely speculative and needs to be justified much better.

Reply: The ozone double-layer structure (stratospheric ozone layer and secondary ozone layer around 90km) is observed in the WACCM-X(SD) ozone anomalies that form at the onset of the SSW and last for about two weeks, resulting in two layers of substantial UV heating. The superposition of these two diurnal tidal waves at the mesosphere may effectively amplify the SDT at high latitudes due to a 12-hour phase offset caused by the different vertical distances both waves have to travel, considering the typical vertical wavelengths of 30-50 km for semidiurnal tides at this latitude (Stober et al., 2021b, 2020).

- line 295: it would be great to give a specific altitude down to which the sun is above the horizon at the given latitudes instead of just mesosphere and stratosphere (see also comment above).

We have added a table to show the calculated altitude at these stations.

**Section 4**

- line 311ff: "explores the enhancement of SDT amplitudes at the onset of SSW ... attributed to zonal wind changes and ozone heating at mid- to low latitudes": This statement added to my confusion on what the authors suggest and present; Firstly, I do not see how the study attributed tidal variability in any way (see main

comment); secondly ozone heating at low- to mid-latitudes is not shown in the paper, so this comes somewhat out of nowhere (unless I missed something?). It goes on discussing the STD anomalies during the recovery phase, which I cannot identify from the STD anomalies presented in Fig. 3 (see comment above).

We have added a figure to show the ozone variation at the mid-latitude station Bern (47N). The effect of ozone VMR on the enhancement of SDT at SSW has been clarified in the reply to major comments. Please see.

- line 315ff: also here, I find it hard to follow the arguments by the authors; again a heating rate by ozone is mentioned (0.5K/day), but it is not clear what this value is based on or where it would be valid; further, it is said to be small (with respect to what?), but in the same range than tidal temperature amplitudes (so not small for tidal variability?)

We have added Figures 1 and 14 in the revised manuscript that illustrate the absolute QRS value reaching up to 20 K/day, with a clear diurnal heating cycle in the middle stratosphere. This diurnal forcing cycle is directly related to ozone and can act as a resource for the diurnal tide in the mesosphere. We provide more details in the discussion.

- line 320: again, which time period is referred to for which STD is enhanced by 40%

Reply: SDT shows a positive anomaly of 10 m/s, with changes reaching up to 40% at the onset of SSWs.

- line 322: I agree that radiative effects are more important in the time following the SSW, but for what? For the mean temperature anomalies or the tidal amplitudes? (For the latter, this is not backed up by results).

Reply: Shortwave radiative heating rate is mainly caused by the ozone absorption of UV in the stratosphere, which reveals the diurnal forcing. Additionally, the double-ozone layer (the stratosphere ozone layer and secondary ozone layer) that forms at the onset of the SSW and lasts for about two weeks, results in two layers of substantial UV heating. The superposition of these two diurnal tidal waves at the mesosphere may effectively amplify the SDT at high latitudes due to a 12-hour phase offset caused by the different vertical distances both waves have to travel, considering the typical vertical wavelengths of 30-50 km for semidiurnal tides at this latitude. This cooling of the middle atmosphere defines the large-scale temperature field and the stratification of the middle atmosphere between the polar and middle latitudes drives the circulation and provides the background condition for the propagation of planetary, tidal, and gravity waves.

- line 325ff: the discussion of the DT is much better to follow and to comprehend compared to the discussion on STD.

Reply: The ozone data from the primary and secondary ozone layer shows clear diurnal cycles, which result in a SDT at the MLT and a diurnal tide at the stratosphere/lower mesosphere. Please see the discussion in the revised manuscript.

**Section 5**

- line 355: I disagree that a deeper understanding on mechanism is provided (see main comment 1)

Changed: This study provides a comprehensive quantification of tidal variability and co-located tracer variability by combining observational data with model simulations and discusses the radiative effects of tracer anomalies on mesospheric tidal variability during SSWs.

- line 361-362: the altered background presumably affects propagation, but this not quantified here. Please clarify that this is a proposed mechanism, rather than a result of the study.

Clarified

---

## Author Response (AR2)

**Response to referee#1's comments for the manuscript**

**General Comment:**

I apologize if my initial requests for major revisions were too much to be reasonably incorporated within the time frame given to the authors.

**General reply:**

We thank the reviewer for the honest words. We take this statement as motivation to learn and deepen our understanding.

**Comment:**

My original concern with the article was that it did not sufficiently quantify the contributions of the different drivers of tidal variability, including that of radiative effects, even though this is central to the paper's theme. I would argue that this concern largely stands also in the revised manuscript. For example, the authors write in the conclusion that "The observed SDT amplification is primarily attributed to the ozone VMR dynamical effect and modified by radiative heating effects modulated by ozone variability", while also arguing that the results of this work provide insights in the coupling between trace gas variations, radiative heating, and tidal variability. Without dedicated analysis quantifying the exact radiative tidal forcing perturbation terms, as well as the actual tidal wind perturbation that would result from these perturbations, this coupling is not identified but is only alluded to. In my view, the presented evidence is not strong enough to provide credibility to the proposed mechanism.

**Reply:** The raised concerns are mitigated by adding a detailed description on how WACCM-X QRS and QRL fluxes are calculated. We also added in the discussion that ozone and water vapor are considered proxies for its heating and cooling rates. The exothermic heating at the MLT due to atomic oxygen is still modulated by the diurnal variation of ozone. We add this in more detail to the revised manuscript.

Replies to a select few of the author's replies to reviewer comments are given below.

**Comment:**

Author Reply: The SSW events are already presented in a table in our paper, which we have cited accordingly. In this study, we focus specifically on major SSWs that occur in mid-winter (January and February) and do not consider events with onsets such as March 24th, 2010. Generally, SSWs occurring in early spring are referred to as final stratospheric warming (FSW) events rather than major SSWs. Additionally, we confirm that the reference model from the cited NOAA page is MERRA-2. Following the list of events that the paper refers to, the 24th of March 2010 is classified as a major SSW. A major SSW in March does not necessarily imply that it was a final warming, even if in this case this may have been so. Indeed the list of major SSW dates includes the definition that the events are only counted if they occur before the 10th of April. I further fail to see why this being a final warming would fundamentally change the relevance of the proposed mechanism. On the contrary, due to the much higher abundance of short-wave radiation in March, I would expect the March event to be a prime example of the proposed mechanism.

**Reply:** There is indeed a significant difference between very late SSWs and those in January and February. Tertiary and secondary ozone are quiet different in March and April. However, the proposed effect requires multi-layer heating to get some kind of mixing effect. Climatologies of the tertiary ozone VMR show that this layer disappears in March.

**Comment:**

It was not clear that the modeled SSW analysis is based on only 4 simulated SSW events spread over 9 years of simulation data. With this low number of event events and simulation years, statistical significance should be discussed. Basically half the data set now contains a major SSW in January/February. Surely this must cause contamination of the climatological model average from which the anomalies are calculated? Especially when looking at such long periods before and after the central SSW dates (spanning 130 days in total). SD-WACCMX v2.1 data is freely available for download spanning many decades of simulations, including the short-wave and long-wave radiative heating fields.

**Reply:** A key aspect of this study are the radiometric observations at Ny-Alesund. The WACCM-X data was added to guide the interpretation of the observations and to compare our measurements with the model. Furthermore, we had to perform our own runs, as all other available WACCM and WACCM-X runs, we are aware of, did not contain QRS, QRL, ozone and water vapor with a sufficient temporal resolution to investigate the effects of SSWs. We also analyzed the Specified Dynamics (SD/WACCM-X) simulation run from Gasperini et al., 2020 and an even longer one back to 1980. However, the required fields were only available as monthly mean profiles and, thus, not suitable for this type of study.

**Comment:**

Author Reply: The superposition of these two diurnal tidal waves at the mesosphere may effectively amplify the SDT at high latitudes due to a 12-hour phase offset caused by the different vertical distances both waves have to travel, considering the typical vertical wavelengths of 30-50 km for semidiurnal tides at this latitude Stober et al., 2021b, 2020). From what I can see, the cited works of Stober et al. (2021b, 2020) provide detailed analysis of seasonal semidiurnal tidal characteristics, as well as its response to SSWs. This includes a description of the tides' vertical wavelength. However, as stated in my original review, I still can't see how the sum of two superposed 24 hr waves can (effectively) amplify the STD. The sum of two 24 hr waves is always a 24 hr wave itself. Even if one would imagine two diurnal waves with a 12 hr phase offset overlain only with the STD, the wavelength of the STD covers only half a wavelength of the DT. In time-frequency (Fourier) analysis, the 12 hr and 24 hr wave forms are also orthogonal. If it is implied that a 24 hr waveform (the sum of two 24 hr waves) affects the solution to the 12 hr waveform's amplitude, this sounds like a short-coming of the analysis.

**Reply:** The superposition of two oscillations with the same periods results in a new oscillation of that period. However, waves are vectors and the superposition of two waves can lead to wave mixing. The mixing of waves is given by its product of the amplitude, and the child wave can reach substantially larger amplitudes as both source waves.

**Comment:**

Author Reply: The ozone data from the stratosphere ozone layer and secondary ozone layer show clear diurnal cycles, which result in a semidiurnal tide at the MLT and a diurnal tide at the stratosphere/lower mesosphere. WACCM-X(SD) fields are in good agreement with the trace gas measurements. We have clarified more details in the added case study of SSW in 2018/2019 and the discussion part. QRS and QRS are not shown in K/day and with a temporal resolution of 3 hours to provide a visual evidence for the difference in their diurnal structure.

The visual evidence does not look convincing, as the amplitude of the 24 hr wave forcing (or 12 hr and 8 hr wave forcing) cannot be estimated by eye. The peak daily heating rates might correspond to only a very short period of insolation, therefore not necessarily resulting in a prominent 24 hr wave forcing. As for the first review, I suggest time-frequency (Fourier) analysis of the 3-hourly modeled heating rates to determine the actual amplitudes of the diurnal, semidiurnal and terdiurnal wave forcings.

**Reply:** We explicitly wrote in the discussion that the true heating is likely not 24 hours. However, to furthermore substantiate the excitation of tidal waves in the wind we added a Figure showing the results of the adaptive spectral filter for diurnal and semidiurnal wind perturbations for two selected periods during the SSW 2018/19 above Tromso.

**Final comment:**

Line 151 of the revised manuscript references Liu et al. (2010a) for a description of the radiative transfer scheme used in WACCM-X. I think it is important to mention all relevant (E)UV absorbing trace gases by name also in your paper, since it is the focal point of your work. Liu et al. (2010) Figure 2 further shows that atomic oxygen number densities surpass that of ozone above approximately 65 km altitude. Since atomic oxygen absorbs UV radiation and is important to the energy budget of the upper atmosphere, its importance/relevance to the short-wave radiation perturbations should be discussed. Especially because QRS shows a maximum above 80 km concurrent with reduced ozone variations in Tromsø in Figure 2 of the revised manuscript.

**Reply:**

We added explicitly the importance of atomic oxygen and exothermic heating in the MLT region in the revised manuscript. Atomic oxygen is still closely related to the diurnal cycle of ozone and, thus, the total heating from ozone, water vapor and atomic oxygen still would reflect a diurnal pattern. We explicitly wrote in the discussion that ozone and water vapor are only proxies for the heating and cooling rates and other species contribute as well.

**Response to referee#2's comments for the manuscript**

The revision of the manuscript by Shi et al. addressed most comments and concerns, and the revised manuscript is overall improved. It is stated more clearly that the causality between tracer anomalies and tides is not yet proved by the analysis, the inconsistencies in the description of the STD tide anomalies have been clarified (even though I still find the proposed mechanism very speculative, see below), as well as the methodological description. However, there are still some parts that I find hard to follow (e.g., in the discussion). I appreciate the work going into the addition of the case study, but have to admit I do not see how this helps to clarify the raised concerns, or how it adds value to the paper. Overall, I recommend minor revisions, with specific comments as detailed in the following.

**Main comments:**

- General comment on Section 3.1., case study, and on connection of QRS and ozone: While I appreciate the addition of the case study, I have to admit that I do not quite see the additional value. In the response, one of the reasons given for adding the case study is that it reveals the diurnal cycle in the QRS forcing more clearly - however, this is also apparent in the added Figure 14 of the composite analysis. Indeed, in the case study the connection of the ozone variations and QRS is less clear to me - ozone shows a clear diurnal cycle also at high altitudes (80 to 100 km), while QRS does not. If ozone is the primary forcer for QRS, this seems inconsistent.

Another reason for adding the case study is given in response to my comment on adding significance tests to the tidal anomalies (previous comment on line 191); the authors reply that the case study reveals significant anomalies, but I do not see that a significance test has been performed here. If anything, the tidal anomalies of the case studies appear much more noisy compared to the composite, so I would rather think the opposite is true (the case study is less significant). I still would recommend the authors to perform a significance test (probably a simple T-test is sufficient, testing whether the anomalies are different from zero) on the anomalies of the composites, which would add value to the study. One reason I could see for adding the case study is the detailed comparison of WACCM-X results and MR+MERRA - if so, I recommend to state this more clearly and to focus on this part.

Reply: The primary motivation for including the case study is to provide a detailed comparison between WACCM-X/SD and observational datasets from MR and reanalysis data MERRA-2 during an individual SSW event. Meteor radar measurements more accurately capture mesospheric winds and tidal amplitudes. WACCM-X/SD complements these observations by offering a more comprehensive representation of the vertical structure of the stratospheric and secondary ozone layers, particularly the clear diurnal cycle of shortwave radiative heating (QRS) in the stratosphere during SSW, which is essential for understanding the proposed forcing mechanisms. The case study helps interpret the subsequent composite analysis of tidal and trace gas anomalies. The revised manuscript now clearly distinguishes the stratospheric ozone layer and the secondary ozone layer (around 90 km). In the stratosphere, QRS is dominated by short-wave ozone heating, leading diurnal cycle. In the MLT regions, we added explicitly the importance of atomic oxygen and exothermic heating at high altitudes (80 to 100

km) in the revised manuscript. Atomic oxygen is still closely related to the diurnal cycle of ozone and, thus, the total heating from ozone, water vapor and atomic oxygen still would reflect a diurnal pattern. A new schematic (Fig. 18) illustrates how mixing of two diurnal heating waves can propagate and amplify the enhanced semidiurnal variability via wave-mixing.

- Mechanism of STD enhancement: I can follow now the reasoning by the authors, in that they propose that the strong STD enhancement just after the SSW onset is connected to a superposition of DT generated by shortwave forcing (in turn driven by ozone anomalies) in the stratosphere and MLT, and that a phase shift in the upward propagation of the stratospherically DT leads to the STD anomaly higher up. This is an interesting mechanism, but if the authors want to go beyond speculating on this mechanism, additional analysis would be necessary. For example, the 12h phase shift is argued based on typical wavelength, and on Fig. 1 (see line 440). I agree there is some phase shift apparent in Fig. 1, but it is nearly impossible to clearly identify the amount of phase shift by eye in this Figure. This argument could be strengthened by explicitly analyzing the phase shift. Furthermore, if this superosition was the case, wouldn't one expect an anomaly in the DT in the stratosphere around the same time than the STD anomaly? However, the DT only sets in after day ~20 (Fig. 8), which is consistent with the strengthening QRS anomalies in the stratosphere around this time. As detailed in the following, I recommend the authors to clearly state what is shown in the paper, and which connections or mechanisms are rather speculative.

Reply: We appreciate the reviewer's thoughtful assessment of our proposed mechanism regarding the enhancement of the semidiurnal tide following the SSW onset. We hypothesize that this enhancement may result from the mixing of diurnal tidal waves generated by QRS, mainly resulting from ozone in the stratosphere and ozone, atomic oxygen, and other chemical species in the mesosphere/lower thermosphere. The mixing of both diurnal tides could potentially lead to wave mixing between both diurnal tides that amplify the semidiurnal amplitude. In the revised manuscript, we will clarify the phase shift. Depending on the vertical wavelength of the stratospheric diurnal tide, there is a certain phase offset of the arrival time between the MLT radiatively forced diurnal tide and the one propagating from below. We added a Figure showing the results of the adaptive spectral filter for diurnal and semidiurnal wind perturbations for two selected periods during the SSW 2018/19 above Tromsø. Regarding the timing of the DT and STD anomalies, we will address this apparent discrepancy more explicitly and propose that the spacing between the layers of absorption, as well as the effective vertical wavelength of the diurnal tide play a crucial role in the SW2 enhancement.

We performed a spectral filtering of the QRS to extract the DW1 and SW2 modulation and overlayed the filtered QRS over the winds and temperatures from WACCM-X(SD). We added temperature to illustrate that these tides propagate upward and that the magnitude of the excitation from the QRS is approximately reasonable for the temperature tidal amplitude. The semidiurnal modulation of the QRS at the mesosphere (80-90km) shows a clear enhancement of the semidiurnal modulation at the time of the enhancement around the SSW. We just present it as a diagnostic. We won't claim that the QRS is the sole driver, but apparently, the QRS semidiurnal modulation leads the enhancement.

**Minor comments:**

- line 189: can you specify how the MR + MERRA product was merged? Is it simply MERRA below a certain altitude, and MR above, or is there a transition?

Reply: The MR + MERRA product uses MERRA-2 data below a certain altitude (white horizontal line), and meteor radar (MR) observations above. There is no transition.

- line 193: can you specify what is mean by "overlapping region" here?

Changed: Zonal winds agree within 2–5 m/s in the altitude region around 75 km, where both MERRA-2 and MR provide valid data and their coverage largely overlaps.

- line 212: given the QRL is tightly coupled to temperature, showing temperature anomalies could be helpful (same applies to composite Figures)

Added the figures in the appendix.

- line 214: Is the statement, that QRL seems to be partly driven by water vapour changes based on the spatial coherence? Given that in the next sentence, it is argued that it is unlikely that water vapour is the main driver, please consider re-formulating to something like: "These changes in the cooling are aligned with water vapour anomalies, but given the mean cooling rate..."
Changed

- line 218: "For the short-term...": this is what will be argued in the paper in the following, rather than a fact, is it? I recommend removing the sentence, or changing it to "might play a role"; Or if this is meant as a fact, add a reference that proofs this connection.

Changed

- line 220: are the anomalies of the secondary ozone layer really caused by transport ("intrusion and exchange of air masses"), or rather by changes in the local chemistry, e.g. due to temperature? Given the diurnal cycle in ozone, i.e., short lifetime, the latter appears more plausible to me.

Changed: Furthermore, the primary driver of disturbances in the secondary ozone layer is more likely related to temperature-dependent chemical processes. Atomic oxygen is still closely related to the diurnal cycle of ozone and, thus, the total heating from ozone, water vapor and atomic oxygen still would reflect a diurnal pattern.

- line 241: I personally prefer the term "composite" over "superposed epoch", but this might be a matter of taste and/or differences in disciplines which term is more common (in stratosphere/troposphere analysis of SSWs, the term "composite" is commonly used).

Changed

- line 328: remove "reduced" before water vapour, as there are both positive as well as negative anomalies.

Changed
- caption Fig. 12: should read ozone, not water vapour.

Changed

- line 371ff: discussion on impact on tides: I recommend moving this to the next section, where the heating rates are discussed and shown.

Changed

- line 376ff (starting with "With the polar wind ..."): following up a previous comment, I still do not see why those sentences on ozone at low and mid-latitudes are added here - is this meant to point out that other studies have found signals at other latitudes (if so, re-write to make this clear), or as comparison to results shown here? If the latter, it might be good to move those sentences to Section 4, where the additional analysis from a mid-latitude station is shown. Another potion would be to simply remove the sentences.

Removed

- line 392: The QRS heating might not be exclusively driven by ozone (see e.g. line 517), so please remove the bracket, or add "(including effects by ozone)".

Changed

- line 394: "reduced heating": The anomalies are still positive, but not as strong, correct? Please rephrase.

Changed

- line 395: "captures the diurnal cycle": I can only see the diurnal cycle at lower altitudes in the stratosphere, but not in the MLT, where primary anomalies are found. Please clarify.

Clarified: Furthermore, the QRS shows a good correspondence to the ozone VMR and captures the diurnal cycle in the stratosphere.

- line 401 / Fig. 14: Overlaying contours of temperature (anomalies) might be helpful in the discussion of the strong connection of longwave cooling and temperatures.

Changed

- line 420: please clarify at which altitudes the QRS anomalies occur during which times - in the MLT around the SSW onset, and in the stratosphere 20-60 days later, correct?

Clarified: The largest QRS values are observed after the central day of the SSW at the MLT, and subsequently in the stratosphere about 20 to 60 days later, when the elevated stratopause reaches again the typical stratospheric altitudes.

- line 449: please refer the reader to the appendix for the tide anomalies in WACCM-X.

Changed

- line 457: Here would be a good place to state that the delay in the QRS (and thus DT) response in the stratosphere is related to the increasing sunlight over the season. This makes for a good transition to the following paragraph on the illumination heights.

Added: The delayed response of QRS (and consequently DT) in the stratosphere is likely associated with the seasonal increase in solar illumination, as sunlight progressively reaches lower altitudes.

- line 501: I recommend removing this part on longwave heating (starting with "However,.."), as it adds no new insights revealed by the study.

Reply: Longwave cooling rates play the crucial role of the background condition for the propagation of tidal waves by analyzing the dynamical effect term, as well as shortwave heating rates.

- line 510: not sure what " a proxy of the changes" refers to - please clarify.

Changed: a proxy of the radiative heating changes

- line 517: this is an important information, that was mostly omitted in the discussion of the results so far. Please check the previous sections to state more clearly that in the MLT, not only ozone is influencing shortwave heating rates.

Reply: We added explicitly the importance of atomic oxygen and exothermic heating in the MLT region in the revised manuscript. Atomic oxygen is still closely related to the diurnal cycle of ozone and, thus, the total heating from ozone, water vapor and atomic oxygen still would reflect a diurnal pattern. We explicitly wrote in the discussion that ozone and water vapor are only proxies for the heating and cooling rates and other species contribute as well.

- line 519: as discussed previously, longwave radiation is mostly a function of temperature, and the effects of water vapour cannot easily extracted (and might be small above the lower stratosphere). Thus, I recommend to remove the sentences starting with "The longwave cooling...", as it adds no new information.

Removed

- line 538ff: I appreciate the effort of adding the additional stations at mid-latitudes. However, I have to admit it is not clear to me what this analysis adds for the conclusions of the paper. Please clarify.

Reply: The mid-latitude observations are used to map the latitude–dependent SSW signatures and to identify the transition zone where dynamical and chemical effects of SSWs fade. Collm (51.3° N) still lies near the polar-vortex edge and therefore retains weakened—but detectable—tidal anomalies, whereas Zimmerwald/Bern (47° N) shows no obvious ozone and water vapor response to SSWs. The contrast demonstrates that (i) the high-latitude stratospheric ozone increase and the short-wave radiative heating (QRS) that are associated with the enhancement of the diurnal tide (DT) after SSW onset (20-50 days) and (ii) Applying the same epoch analysis to the ozone and water vapor measurements from GROMOS and MIAWARA at midlatitudes reveal no discernible signatures associated with the SSW events. Therefore, the positive ozone anomalies in the stratosphere and MLT at high latitudes act as radiative drivers that accompany upward-propagating diurnal waves, forcing stronger SDT amplitudes in the MLT at the onset of SSW.

- line 584: Please consider re-phrasing to "... driven by stratospheric ozone anomalies, whose radiative impacts become effective 20 -30 days after the SSW onset due to the strengthening solar radiation over the season."

Changed

- line 587: no attribution to dynamical effects was presented here, so consider re-phrasing to "... is likely primarily due to dynamical effects, ..."

Changed

- line 590: see comment above, please clarify here that this is the proposed mechanism that needs to be shown in future studies.

Added